# Carbon Dioxide Concentration Levels and Thermal Comfort in Primary School Classrooms: What Pupils and Teachers Do

Maria Gabriela Zapata-Lancaster *, Miltiadis Ionas, Oluyemi Toyinbo and Thomas Aneurin Smith

Welsh School of Architecture and School of Geography and Planning, Cardiff University, Cardiff CF10 3NB, UK
* Correspondence: zapatag@cardiff.ac.uk

**Abstract:** The current climate emergency concerns and the COVID-19 pandemic demand urgent action to maintain healthy indoor environments in energy efficient ways. Promoting good indoor environments, in particular, increasing ventilation levels, has been a prominent strategy to mitigate the risk of COVID-19 transmission indoors. However, this strategy could be detrimental to thermal comfort, particularly during the heating season in buildings located in temperate climate zones. This paper presents research conducted in two primary schools in South Wales (UK) where the temperature, relative humidity and the carbon dioxide ($CO_2$) concentration levels were monitored. The study monitored six classrooms and two communal spaces in the two schools during the academic year 2021/2022, the first academic year back to teaching and learning in school buildings after home-schooling and educational disruptions due to COVID-19 lockdowns. The study investigated the actions taken by teachers and pupils to balance the thermal comfort needs while minimising $CO_2$ concentration levels. We conducted user studies to explore the comfort perceptions by pupils and teachers in relation to the thermal conditions and the freshness of air in the monitored classrooms. The paper identifies opportunities where end-users, teachers and pupils engaged with the management of the indoor environmental conditions and adopted actions to balance the requirement of reducing $CO_2$ concentration levels while promoting thermal comfort. This research offers lessons and insights related to end-users' agency and their understanding of indoor environments and thermal experience in schools.

**Keywords:** $CO_2$ concentration indoors; thermal comfort; school buildings; monitoring



## 1. Introduction

There is an urgent need to promote healthy and comfortable indoor environments in energy efficient ways as means to respond to climate emergency and health concerns. Ensuring good indoor environmental quality (IEQ) is paramount to promote the health and wellbeing of occupants. Research has found that IEQ impacts the health, wellbeing and performance of building occupants [1–4]. School children spend significant proportion of their time in school buildings. Given their physiology, children are likely to be more vulnerable than adults to poor indoor conditions, including thermal environments and carbon dioxide ($CO_2$) concentration levels [5–7]. Therefore, promoting healthy and comfortable classroom conditions is fundamental to ensure the provision of spaces where children can learn, play, socialise and thrive in school buildings [8].

This research explores two key aspects of indoor environmental conditions: thermal conditions and $CO_2$ concentration levels. Previous thermal comfort research that focused on school settings applied monitoring studies to identify the existing indoor conditions [9], by explicitly investigating the difference between thermal comfort preferences and perceptions between children and adults [10,11]. Teli et al. [12] analysed the thermal perception and preferences of primary school children aged 7–11 via questionnaires, exploring overheating risks in schools as perceived by children in comparison to the definitions of overheating as per standards ISO 7730 [13] and EN 15251 [14]. Their work finds that young children

may experience higher risk of overheating than typical thermal comfort guidance suggests. Research investigating the nexus between $CO_2$ concentration levels in schools and children's health and wellbeing find that $CO_2$ concentration levels above 1000 ppm can result in decreased levels of attendance [15]; whilst it can also increase the number of dry cough and rhinitis cases in school children [16] as well exacerbate the risk of asthma [17]. It has been found that when $CO_2$ levels were measured above 1500 ppm; there are increased number of errors in performance tests compared to 900 ppm [18]; increased difficulty to concentrate [19], and a reduction in annual school attendance by pupils [20]. Industry guidance and standards such as Building Bulletin 101 [21], CIBSE Guide A [22] and EN 16798-1 [23] provide recommendations related to $CO_2$ concentration levels and thermal comfort in school buildings, identifying different categories of indoor environment quality. Building Bulleting 101 [21] recommends that $CO_2$ concentration levels in school buildings are kept below 1000 ppm as an indicative benchmark of good indoor air quality (IAQ). Acceptable levels are between 1200 and 1500 ppm while $CO_2$ concentrations above 1750 ppm flag up the need of additional ventilation.

Historically, the drive to reduce building energy use associated to heating in buildings located in temperate climate zones has led to the design and operation of buildings where ventilation rates are reduced to prevent heat losses [24]. However, this strategy may be problematic in the light of post-COVID-19 requirement that promote increased ventilation levels. The COVID-19 pandemic has encouraged people to minimise $CO_2$ concentration levels indoors. Governments have been proactive to issue guidance to promote well-ventilated schools for health and safety purposes. For example, Welsh Government guidance recommends schools in Wales to comply with health and safety guidance and undertake proportionate control measures to mitigate COVID-19 risks. The 'Operational guidance for schools and settings to support limited attendance' issued in 2020 calls for schools to measure $CO_2$ levels [25]. In October 2021, the Welsh Government started a programme to distribute $CO_2$ monitors in educational settings, outlining how the monitors should be used in the document 'Carbon dioxide monitors in educational settings' [26]. The guidance uses a traffic light system to classify $CO_2$ levels as per ventilation quality: green light (400–800 ppm) signals adequate ventilation; amber light (800–1500 ppm) suggests inadequate ventilation and improvement needed; and red light (above 1500 ppm) shows poor ventilation with action required. Schools had to respond quickly to COVID-19, proposing actions and strategies to mitigate the risks of education disruption, and to promote ventilation in classrooms and the health and safety of pupils and teachers.

This research investigated the $CO_2$ concentration levels, the temperature and the relative humidity in a small number of classrooms in two primary schools located in South Wales, United Kingdom during the 2021/22 academic year (September 2021 to July 2022), the first full academic year where pupils returned to schools after lockdowns and home schooling. The research conducted user studies to identify the teachers' and pupils' satisfaction levels with the ventilation and thermal conditions that they experienced in their classrooms and explored the actions and behavioural changes adopted to enhance the ventilation in the classrooms while maintaining thermal comfort. The aim of the paper is to identify the pupils' and teachers' satisfaction with the temperature and the air freshness in their classrooms in relation to monitoring data and the actions adopted in response to COVID-19 guidance. We consider that the rapid adaptation and responses of schools, teachers and pupils to COVID-19 highlight opportunities to tap into the interest of building end-users to interact with and manage their indoor environments, to encourage healthy, comfortable and sustainable buildings. This contests the notion that building end-users are inherently passive, disengaged, ill-motivated and deficient in knowledge to interact with buildings and their technologies to achieve building performance goals and address wider societal concerns, such as COVID-19 and climate emergency, through the ways they use and manage buildings.

## 2. Materials and Methods

This research conducted long-term monitoring studies of thermal conditions and $CO_2$ concentration levels in primary school classrooms, and investigated pupils and teachers experiences of thermal (dis)satisfaction in different seasons after the return to school post-COVID-19 pandemic, reporting on results found for the first full academic year back to learning in school buildings in Wales (academic year 2021/22). This research investigated the thermal environment and $CO_2$ concentration levels in two primary schools in South Wales with Display Energy Certificate (DEC) ratings of D, which constitute the most common DEC rating among primary schools in Wales (40% of Welsh schools are DEC D); condition grade A is the grade of 57% of primary schools in Wales; and, suitability grading A is the suitability grading of 81% of primary schools in Wales. Condition grade A is typically a building in very good condition, comparable to new that do not require renovation. Suitability category A defines good facilities suitable for teaching, learning and wellbeing in school (School Data Condition Survey in Wales, 2022). The schools involved in this study, therefore, represent the most predominant DEC rating, condition grades and suitability condition grades of primary schools in Wales. Other key features of the case studies, such as floor area and number of students are summarised in Table 1. In terms of the heating and ventilation services of the school buildings, both schools have gas boilers. School A has radiators and schools B has underfloor heating. None of the school buildings has mechanical ventilation, so they rely on natural ventilation. The chosen classrooms rely on cross ventilation through windows. For classrooms in both schools, at one side there are windows facing the courtyard, and at the other side the classroom door leading to spacious communal spaces (foyer at school A and corridors at school B). The dimensions of the classrooms in school A are: classroom 1: 7.0 m × 8.5 m; classroom 2: 7.5 m × 7.9 m; classroom 3: 7.3 m × 7.8 m, and in school B: classroom 1: 6.1 m × 8.9 m; classroom 2: 6.1 m × 8.9 m; classroom 3: 9.1 m × 8.8 m). Each classroom is occupied by 25 to 30 pupils, one teacher and one assistant. The classrooms are occupied from 9 a.m. to 3:30 p.m.

**Table 1.** Key features of the case studies.

| Case Study | School A | School B |
|---|---|---|
| Area ($m^2$) | 1600 | 1744 |
| EPC rating | D | D |
| Number of students | 200 | 250 |
| Year of opening | 2001 | 2010 |
| Condition grade | A | A |
| Suitability grade | A | A |
| Spaces monitored | Three classrooms, dinner hall, reception area | Three classrooms, reading hall, reception area |

In terms of research design, the research included two types of studies: (1) monitoring studies to measure the temperature, relative humidity and $CO_2$ concentration levels in three classrooms per school as a means to characterise the indoor conditions in different seasons during the 2021/22 academic year; and, (2) user studies to explore actions taken to respond to COVID-19 and to identify the pupils' and teachers' satisfaction levels with the thermal conditions and ventilation in their classrooms. The implementation of these two approaches enabled the research to characterise the indoor environmental conditions, with focus on thermal conditions and $CO_2$ concentration levels, in different seasons of the year and in relation to actions and perceived satisfaction levels of pupils and teachers using the monitored spaces.

*2.1. Monitoring Study*

The monitoring study aimed to identify seasonal variations in $CO_2$ concentration levels and thermal conditions in the schools. The $CO_2$ concentration levels, the temperature and the relative humidity of three classrooms and two communal spaces were measured in each of the primary schools in the academic year 2021/22 from September 2021 to July 2022. The monitored classrooms were located in different orientations within the school building. The temperature and relative humidity were measured using ALTA wireless T/RH sensors with an accuracy of $\pm 2\%$ RH (reading rage 0–100% RH) and $\pm 0.5\,^{\circ}$C (reading rage 0–100 $^{\circ}$C). Two wireless temperature and relative humidity sensors were placed in each classroom, at chest height. For validation of the results, two more temperature and relative humidity data loggers (Tinytag) were placed in each classroom. $CO_2$ concentration levels were measured using an ALTA wireless $CO_2$ sensor, with a capability to measure 0 to 10,000 ppm and an accuracy of $\pm 45$ ppm + 3% of reading. The $CO_2$ sensors are designed for ordinary built environments with temperature ranges between 0–50 $^{\circ}$C and humidity 0–95% RH. One wireless sensor was placed in each classroom at a central position of the room, at chest height. Moreover, a second $CO_2$ concentration sensor with a display was placed at a position near the teacher's desk. This sensor was used both for data validation and as an alarm to notify the teacher in case of high $CO_2$ concentration. Finally, there were wireless temperature and relative humidity sensors in two communal spaces of each school, and a particulate matter wireless sensor at the entrance of each school. The photographic evidence include photographical documentation of the monitoring kit installation in both schools. All monitoring devices were set at a 10 min monitoring interval. Both case studies were remotely monitored throughout the entire school year of 2021–22, and there were only minor instances of data loss due to accidental deactivation of the gateway, which was addressed within a few days. The technical features of the sensors used in this study are comparable to sensors used in other studies that have investigated the indoor conditions of buildings, including school building monitoring pre- and post- COVID-19 pandemic (e.g., [27]).

Both schools are located in urban areas that are not densely occupied. All classroom windows face the courtyard, which is spacious, creating a large buffer space between the local environment and the school building. The traffic around the schools is light, and there are no industrial or large commercial buildings nearby. The only possible sources of $CO_2$ emissions are domestic fossil fuel boilers in the area, and the $CO_2$ emissions from the school's gas boiler. The Alta sensors were calibrated based on outdoor measurements of $CO_2$ concentration (400 ppm) and in alignment to manufacturer's calibration instructions. The measurements from the Alta sensors were validated against measurements of two additional $CO_2$ and Temperature Monitors Telaire 7001 (sourced from Tempcon Instrumentation Ltd., West Sussex, UK)

The analysis of the monitoring data presents measurements during occupied hours in weekdays during term time. Data outside occupied hours, weekend data and school holiday data were excluded. The monitoring data are reported in relation to four seasons of the 2021/2022 academic year: (1) Autumn (6 September–22 October 2021); (2) Winter (1 November 2021–18 February 2022); (3) Spring (28 February–8 April 2022), and: (4) Summer (25 April–22 July 2022).

The primary aim of the monitoring study was to identify the existing thermal conditions and $CO_2$ concentration levels in the monitoring spaces to identify seasonal profiles of indoor environmental conditions. The monitored data were analysed in relation to reported actions, adaptations and satisfaction levels of pupils in different seasons.. The monitoring study did not aim to depict a detailed picture of indoor environmental conditions for the entire school, nor to undertake an in-depth investigation of indoor conditions within a space or between different spaces in the participating schools. Instead, it enables the general characterisation of the thermal environment and $CO_2$ concentration level profiles in different seasons and in the light of user studies.

### 2.2. User Studies with Teachers and Pupils

User studies were conducted among school end-users (pupils and teachers) to identify (1) the pupils' and teachers' satisfaction levels with the thermal environment and the freshness of the indoor air; and, (2) the actions taken to foster ventilation and thermal comfort in classrooms in response to COVID-19 ventilation guidance. The user studies with teachers included two types of questionnaires. The first questionnaire explored the teachers' comfort satisfaction with the indoor temperature and freshness of the air in the classrooms where they primarily teach. This questionnaire also explored the seasonal differences in their comfort satisfaction levels and the actions taken to achieve comfort in different seasons. The second questionnaire focused on reporting the actions and adaptations taken in response COVID-19. Teaching staff from all classrooms responded to the questionnaires, with 15 participants taking part in School A and 6 participants in School B, depicting responses for all classrooms within the investigated schools.

The design of the teachers' comfort questionnaire (Appendix A) was informed by standardised building performance surveys and post-occupancy evaluation questionnaires such as the Building User Survey and TOBUS questionnaires [2] that capture respondents' satisfaction levels with various indoor environmental parameters. The questionnaire did not aim to associate the results directly to monitoring data as only three classrooms took part of the monitoring study. The questionnaires aimed to provide a general picture of the indoor conditions in the school as perceived by teachers in different seasons. The teachers' comfort questionnaire had closed-ended questions using five-point Likert scale and free comment boxes to expand the responses to illustrate the teachers' comfort perceptions in different classrooms in the school, including the ones where monitoring studies took place. The second teacher's questionnaire (Appendix B), which focused on actions and adaptations in response to COVID-19, included multiple-choice and open-ended questions with a free comment box. The COVID-19 adaptation questionnaire included questions related to the changes in the use of the school building, spatial modifications and changes in the classroom layout; modifications in the everyday operation and the use of building controls in classrooms; changes in use of the school spaces and the classrooms; and the provision and use of ventilation guidance. The COVID-19 adaptation questionnaire prompted teachers to provide examples of actions adopted to address COVID-19 concerns and to identify various strategies and opportunities for flexible use of teaching spaces in classrooms and outside the classrooms. The design of the COVID-19 adaptation questionnaires was informed by the principles outlined in Bryman [28] to ensure the quality of questionnaire design.

Pupils also took part in the research, sharing their perspectives in relation to their satisfaction levels with the temperature and air freshness in their classrooms and identifying what actions can be taken to foster comfortable classrooms in different seasons. The research engaged with children in Foundation phase (6–8 years old) and children in Key Stage 2 (9–11 years old), approximately 90 pupils per school.

The research instruments used with children generated a mixture of qualitative and quantitative data, and comprised: (1) drawings where pupils expressed their perception of the indoor environment in their classrooms, and actions they take to modify their thermal experience (completed by all pupils participating in the study aged 6–11); (2) questionnaires to children aged 9–11 using traditional thermal comfort surveys tailored to this age group (~100 pupils in 4 classrooms); and (3) child-made films where children filmed parts of the indoor environment in their school and narrated their perceptions of energy use and possibilities to save energy. Children's instruments were administered by the research team during a series of 60–75 min workshops delivered to different year groups in the school under teachers' supervision. Workshops were conducted in November and December 2021, March 2022 and July 2022, and the details of methods and participants involved are summarised in Table 2. Comfort questionnaires were administered to older pupils (aged 9 years old and above) to rate their satisfaction levels with the temperature and air freshness in their classrooms. One questionnaire was administered in November and December 2021 and a second questionnaire was administered in July 2022 to identify satisfaction levels in

the winter and summer seasons. The children's comfort drawings and the questionnaires adopted a 'right here, right now' approach and their responses were considered in relation to monitored data collected in their classrooms at the time of the workshops. It should be noted that there were no additional boundary conditions for the children's comfort surveys (i.e., whether children had eaten before the test, or were excluded due to ill-health), children simply took part if they were in the classroom at the time of testing, and normally occupied that classroom for their everyday lessons.

**Table 2.** Details of research methods and participants involved in pupils' workshops conducted in winter and in summer to explore children's seasonal experiences of indoor environmental conditions in their classrooms.

|  | School A | School B | Research Methods Used |
|---|---|---|---|
| Winter comfort study |  |  |  |
| Date | 6 December 2021 | 29 November 2021 |  |
| Time of workshop with Youngest participants (6–8 yo) | 13.00–14.15 | 9.30–10.30 | Drawing only |
| Time of workshop with oldest participants (>9 yo) | 9.15–10.30 10.45–12.15 | 10.45–11.45 13.30–14.15 | Drawing and comfort questionnaire |
| Summer comfort study |  |  |  |
| Date | 11 July 2022 | 8 July 2022 |  |
| Time of workshop with Youngest participants (6–8 yo) | 13.00–14.15 | 9.30–10.30 | Drawing only |
| Time of workshop with oldest participants (>9 yo) | 9.15–10.30 10.45–12.15 | 10.45–11.45 13.30–14.15 | Drawing and comfort questionnaire |

The design of children's research instruments was informed by recommendations by Christensen and James [29] and by research precedents on children's thermal experience in classrooms [30]. Questionnaires are a predominant method used in thermal comfort research, however, they can be problematic in gathering subjective thermal comfort responses [31], especially when gathering responses from children and young people. Therefore, this study adopted creative methods (drawings and workshop discussion) in combination with children's' questionnaires that were tailored to children's abilities and skills, based on previous evaluations of children's thermal comfort [32,33]). Questionnaires used visual cues, color-coding of the questions, simple concise questions, depiction of pictures and images to represent different responses and ranges of satisfaction levels (see Appendix C). The comfort questionnaires were used in combination with other types of research methods tailored to children's skills and abilities to explore their personal thermal experience in their classrooms [30]. The combination of methods deployed to investigate children's thermal comfort experience responds to the increased recognition that qualitative methods are valuable in gaining richer insights into human dimension of building performance and building use [34], that questionnaires alone may not be able to capture.

The project was approved by the ethics committee of the department leading the research (SREC reference No. 2142), and considered ethical principles and research integrity guidance for research with human participants, in particular children. A package with information about the study, participation involved, mechanisms to request information and to withdraw the study and consent forms was distributed to the headteacher, teachers and pupils' parents and guardians to ensure informed participation and consent. The research team explained the study to teachers during face-to-face and online meetings and also to children during workshops. No personal data from pupils or teachers were collected during the study.

## 3. Results

### 3.1. Monitoring Results

The monitoring data showing the temperature in the classrooms has been illustrated in a figures 'School Average temperatures hourly daily profile' per season. The figures include a shaded rectangle that represents the recommended range of temperature as per CIBSE Guide A [22] and a blue line that shows Building Bulletin 101 [21] recommended temperature. The monitoring data show that the temperature in the classrooms in autumn and winter (Figures 1 and 2), were slightly colder than the recommended guidelines as per Building Bulletin 101 [21] and CIBSE Guide A [22]. Figure 1 illustrates the daily profile of the temperatures in autumn for the three monitored classrooms in School A. The winter daily profile of the temperatures in School A is illustrated in Figure 2. The temperature tends to be slightly colder than recommended by CIBSE Guide A guidance and Building Bulletin 101 (between 19–21 °C). In autumn, the temperature of classroom 3 is over 19 degrees most of the occupied hours. The temperature of classroom 1 tends to remain slightly colder than other monitored classrooms during occupied hours (between 17.5 °C and 18.5 °C). In winter, the average temperature of all classrooms in School A was recorded between 16 °C and 18 °C. The temperature daily profile in spring shows that the indoor temperatures of the monitored classrooms in School A are between 13 and 16 °C (colder than CIBSE Guide A recommendations) while the summer temperatures range is between 20 and 22 °C.

Figure 3 illustrates the temperature profile for three classrooms in School A for the coldest recorded period (Spring 2022) showing the minimum, maximum and average temperatures recorded per hour weekdays during occupied hours. In spring 2022, the minimum temperature recorded was 13 °C at 9 a.m. and the maximum temperature was 16 °C at 3 p.m. Figure 4 illustrates the temperature profile for three classrooms in School A for the hottest recorded period (Summer 2022). In Summer 2022, the minimum temperature recorded was 20 °C at 9 a.m. and the maximum temperature was 22 °C at 1 p.m. Figures 5–8 illustrate the temperature profile for the 3 classrooms in School A in different seasons, showing the minimum, maximum, average temperatures and standard deviations recorded in each season.

In School B, the temperature daily profile illustrated in Figure 9 shows that the temperature in Classrooms 1 and 3 are within CIBSE Guide A recommendations (19–21 °C). Classroom 2 is slightly colder at 18 °C during most occupied times. The temperature daily profile in winter for classrooms 2 and 3 shows that the average temperatures are slightly colder than the CIBSE Guide A recommendation (Figure 10). Only Classroom 1 maintains its average temperature in winter at 19 °C, as per CIBSE Guide A recommendations. Figures 11 and 12 illustrate the temperature profile in spring and summer for the monitored classrooms in School B, showing that temperatures range between 20 and 22 °C in spring and between 21.8 and 23.4 °C in summer. Figures 13–16 summarise the seasonal temperature profile for the three classrooms in School B during the monitoring period, showing the minimum, maximum, average temperatures and standard deviations recorded hourly during occupied weekdays hours.

The monitoring data for relative humidity shows that the average hourly readings of the six monitored classrooms in the two schools remained between 40% and 60% in autumn and winter seasons, in alignment within CIBSE Guide A recommendations.

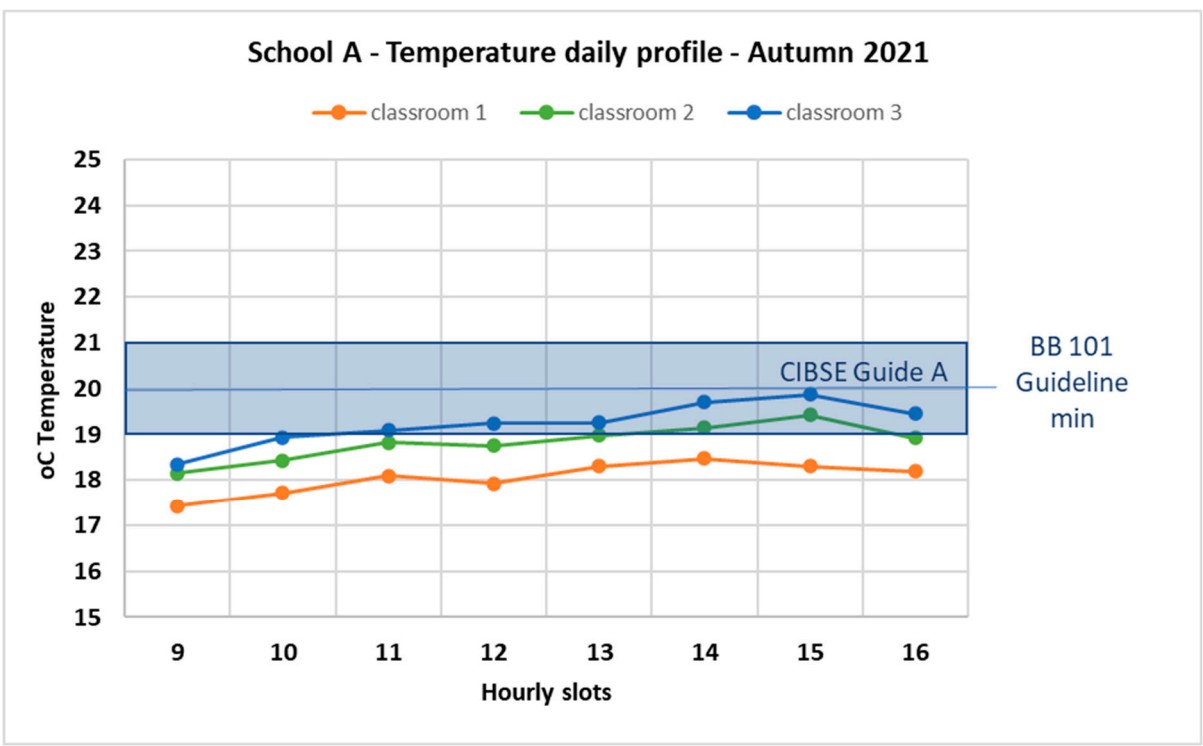

**Figure 1.** School A average temperature, hourly daily profile, autumn 2021.

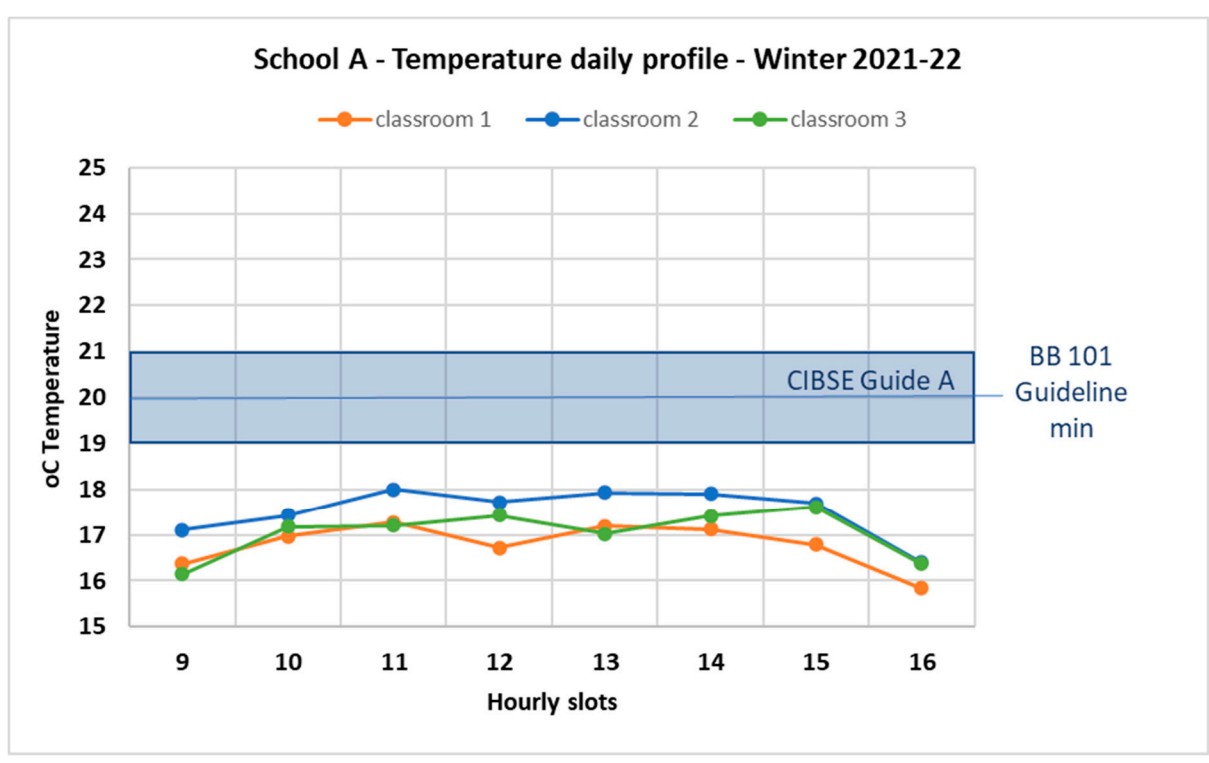

**Figure 2.** School A average temperature, hourly daily profile, winter 2021–22.

$CO_2$ levels are illustrated in Figures 17–24. These figures show the average $CO_2$ levels monitored in the classrooms in relation to Welsh Government guidance [22], which applies a traffic light system to categorise different levels of indoor air quality (green light for $CO_2$ levels between 400 and 800 ppm, amber for $CO_2$ levels between 800 and 1500 ppm and red light for $CO_2$ levels above 1500 ppm). Figure 17 illustrates the average $CO_2$ levels in the three classrooms in School A recorded in Autumn. The $CO_2$ levels tend to remain between 500 ppm and 1000 pm in autumn. Classroom 2 experiences $CO_2$ levels above 800 ppm between 10:00 and 12:00 and between 14:00 and 15:00; Classroom 3's data show that $CO_2$ levels raise up to 1000 ppm between 09:00 and 11:00 and between 13:00 and 15:00. The measured range of $CO_2$ levels in the classrooms in School A corresponds to green and amber zone as per Welsh Government $CO_2$ Guidance. In winter, the $CO_2$ levels are above 800 ppm in Classrooms 2 and 3 (amber zone), with instances of $CO_2$ levels being 1500 ppm at 10.00 in classroom 3 (Figure 18). The $CO_2$ daily profile reflects the occupancy patterns in terms of teaching/learning time in the classrooms as well as the regime of window opening and closing during school time as reported by teachers. In spring, the $CO_2$ levels in classrooms 1 and 2 are predominantly in the green and amber zone during occupied hours, between 800 and 1500 ppm (Figure 19).

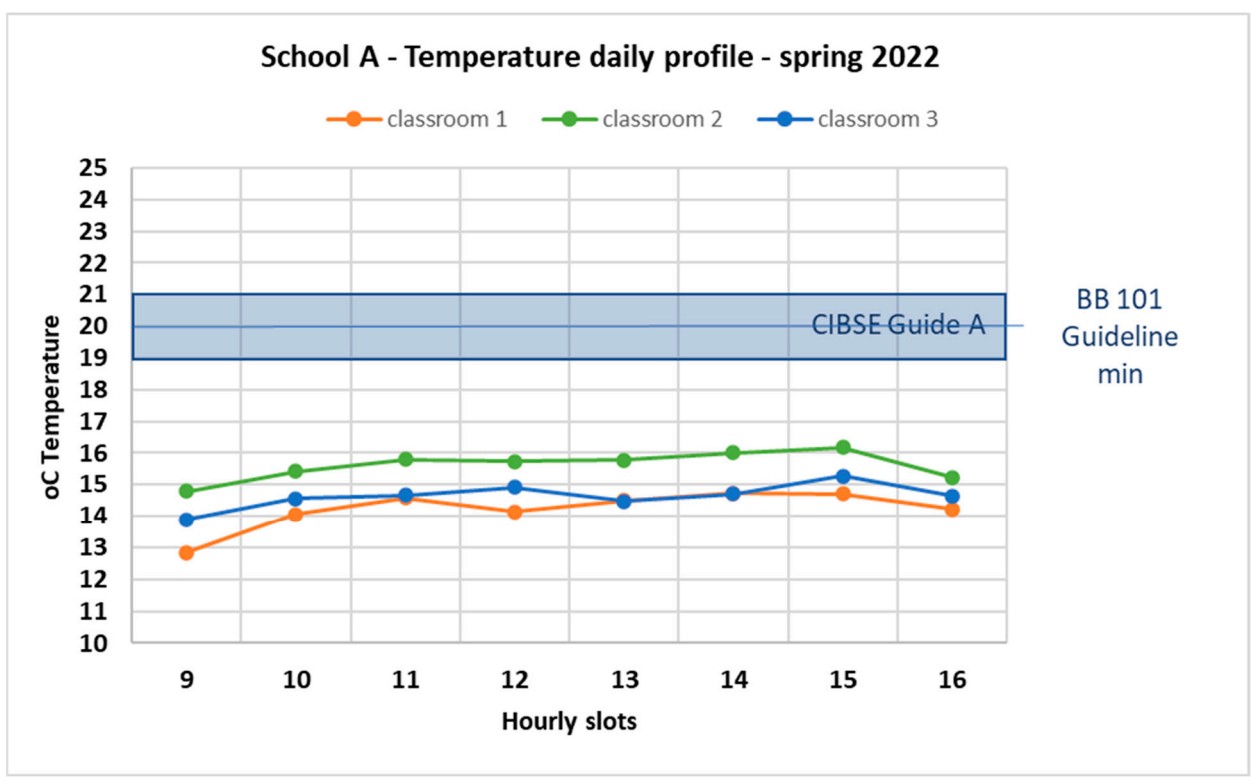

**Figure 3.** School A average temperature, hourly daily profile, spring 2022.

The $CO_2$ levels in three monitored classrooms in School B are within the green zone in autumn as per Welsh Government Guidance, with readings between 500 ppm and 800 ppm (Figure 20). In winter, the $CO_2$ levels in classrooms 2 and 3 were in the amber zone, with levels remaining under 1200 ppm, while classroom 1 had average $CO_2$ levels between 700 ppm (Figure 21).The $CO_2$ levels in the classrooms in School B in spring and summer are illustrated in Figures 22 and 23. For most of the occupied hours, the $CO_2$ levels remain within the green zones in all classrooms (less than 800 ppm), with a maximum $CO_2$ level of 1000 ppm in Classroom 3 between 9.00 and 10.00 in spring.

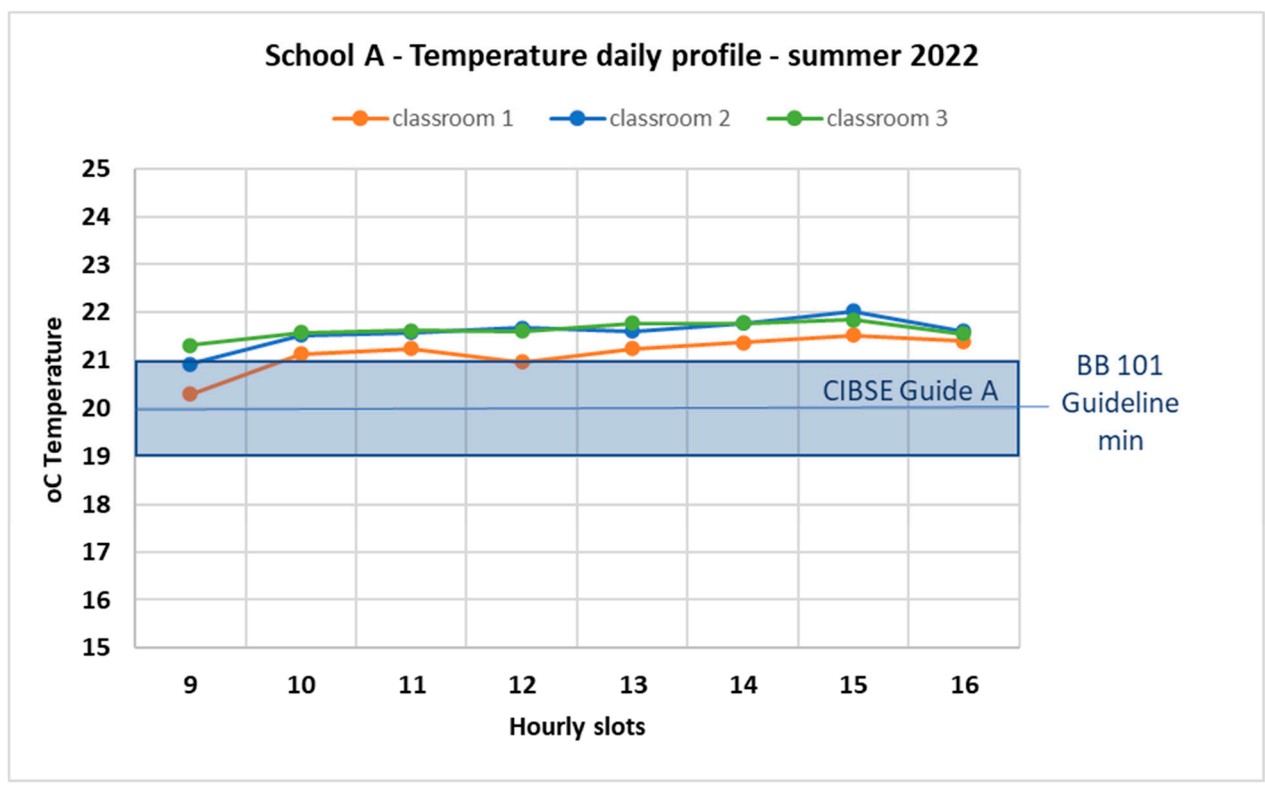

**Figure 4.** School A average temperature, hourly daily profile, summer 2022.

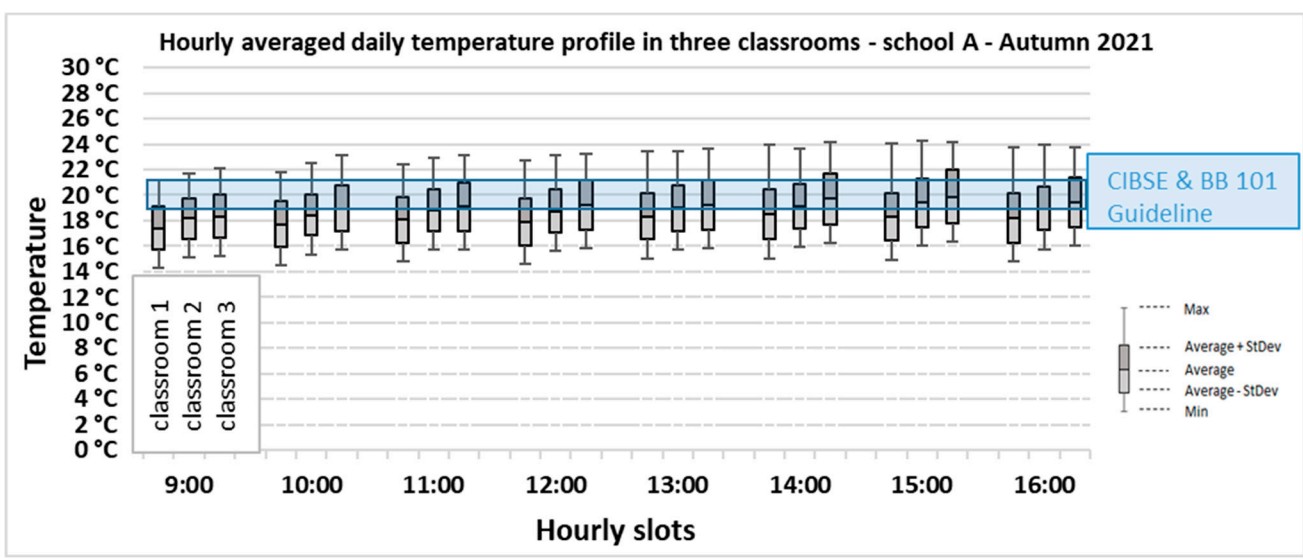

**Figure 5.** School A: hourly averaged daily temperature profile for 3 classrooms showing min, max, average temperatures and standard deviation for autumn 2021–22.

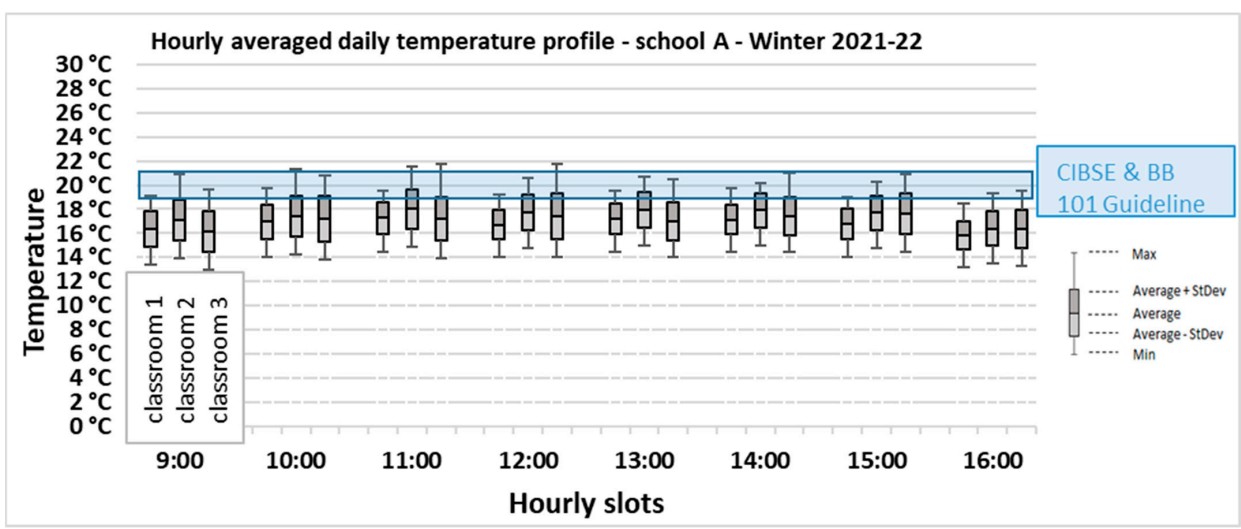

**Figure 6.** School A: hourly averaged daily temperature profile for 3 classrooms showing min, max, average temperatures and standard deviation for winter 2021–22.

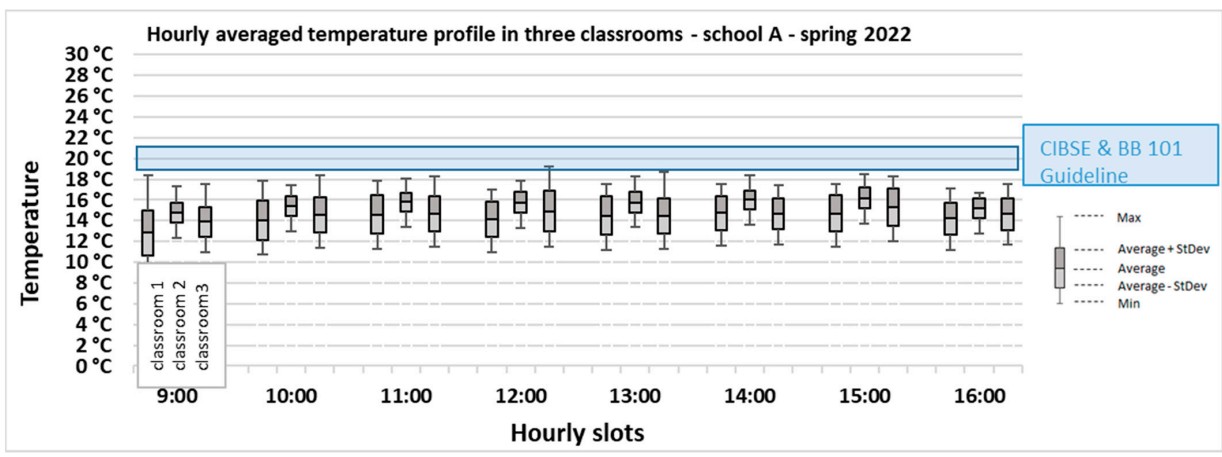

**Figure 7.** School A: hourly averaged daily temperature profile for 3 classrooms showing min, max, average temperatures and standard deviation for spring 2022.

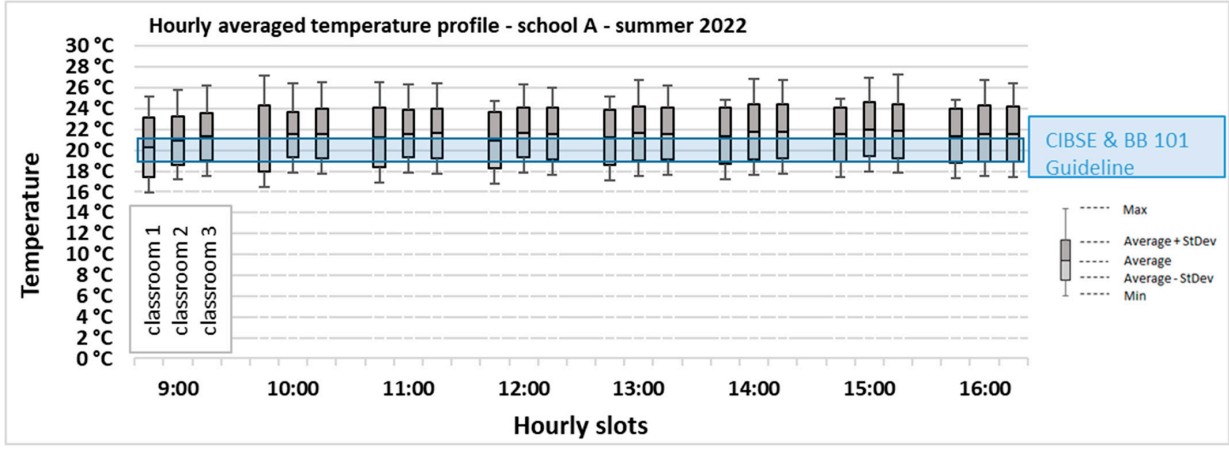

**Figure 8.** School A: hourly averaged daily temperature profile for 3 classrooms showing min, max, average temperatures and standard deviation for summer 2022.

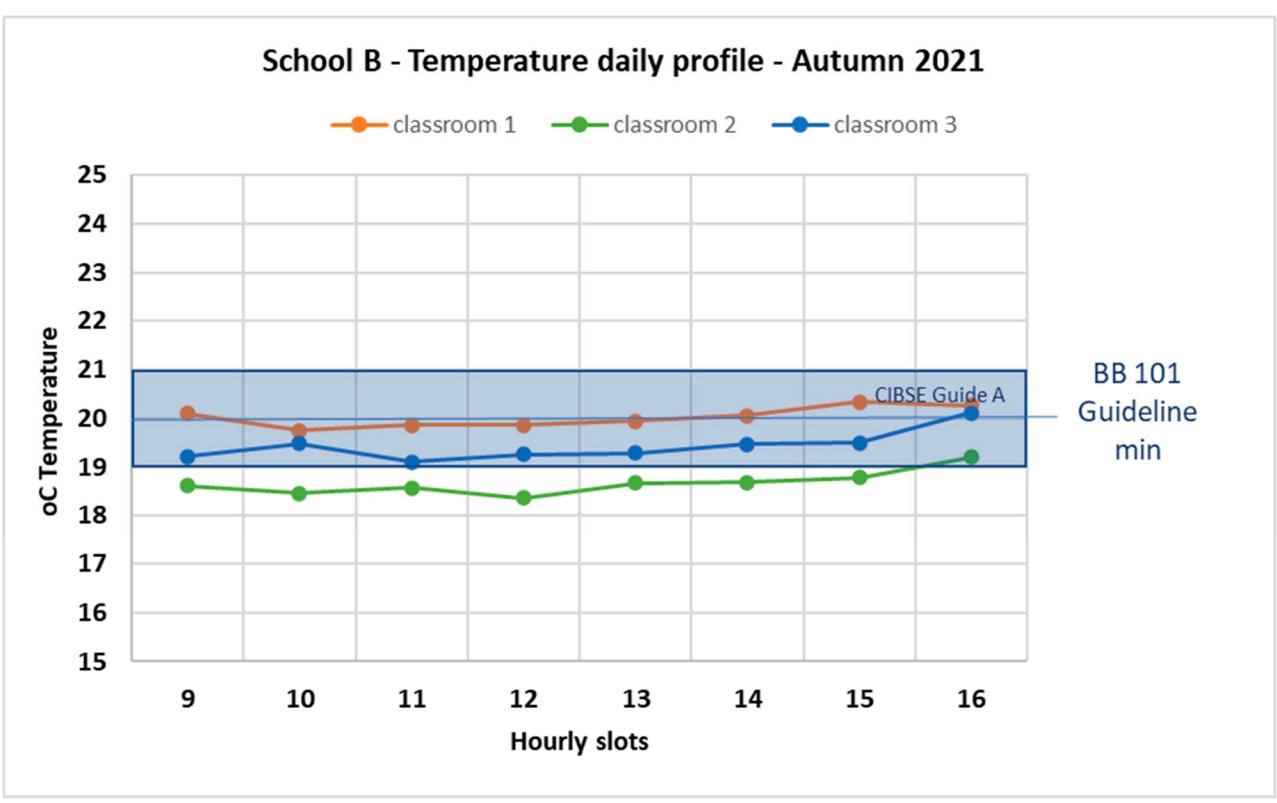

**Figure 9.** School B average temperature, hourly daily profile, Autumn.

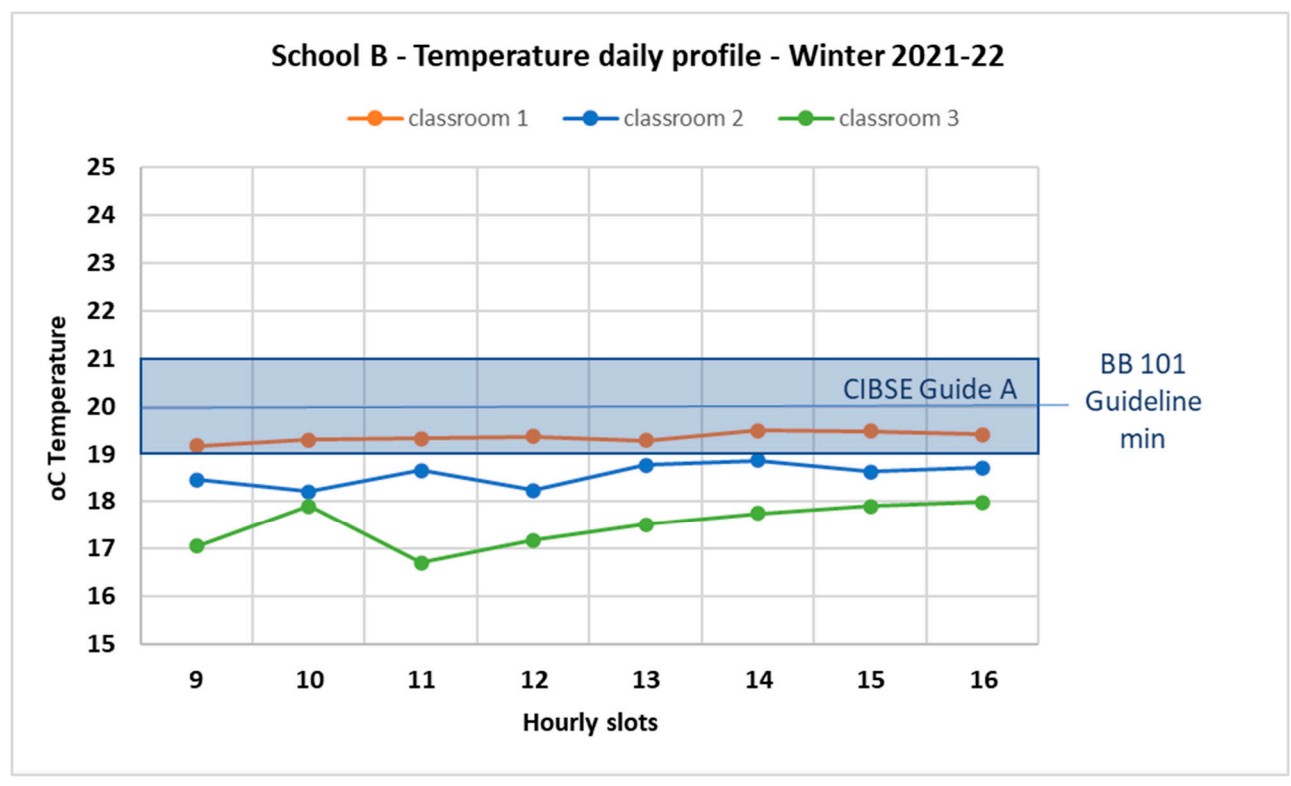

**Figure 10.** School B average temperature, hourly daily profile, winter.

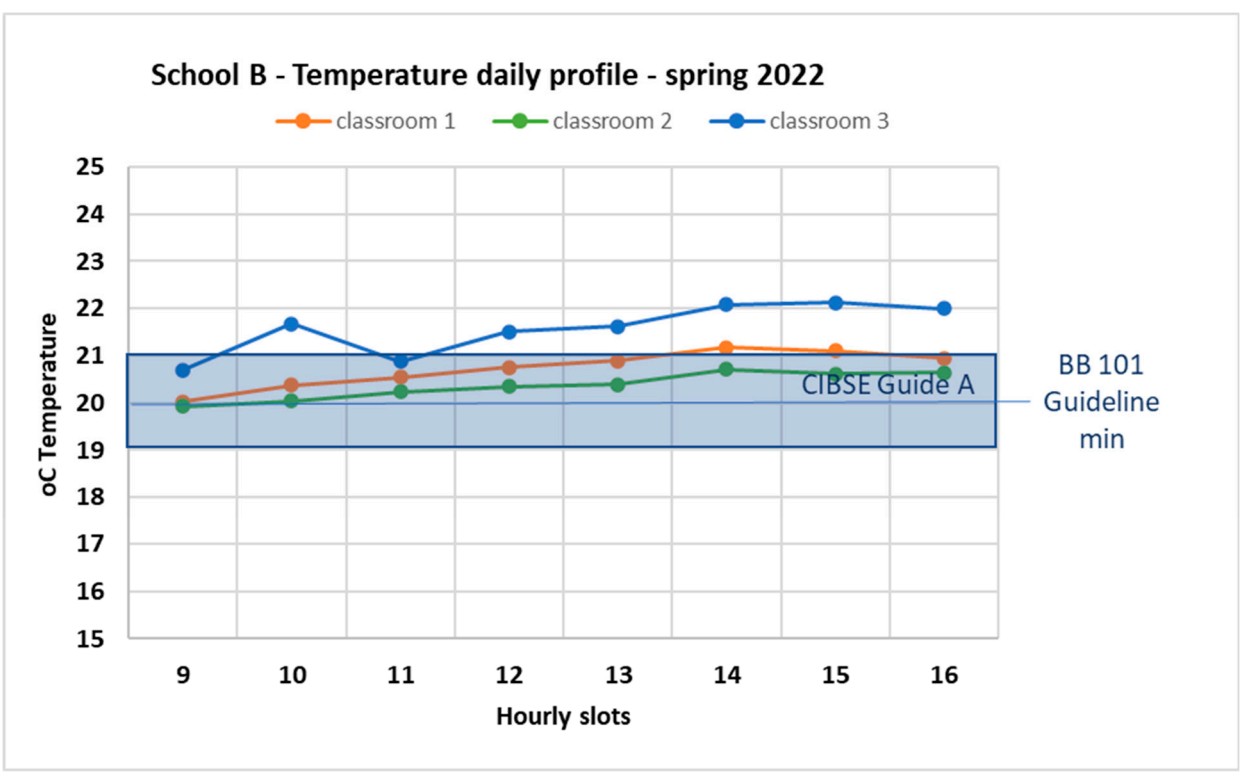

**Figure 11.** School B average temperature, hourly daily profile, spring.

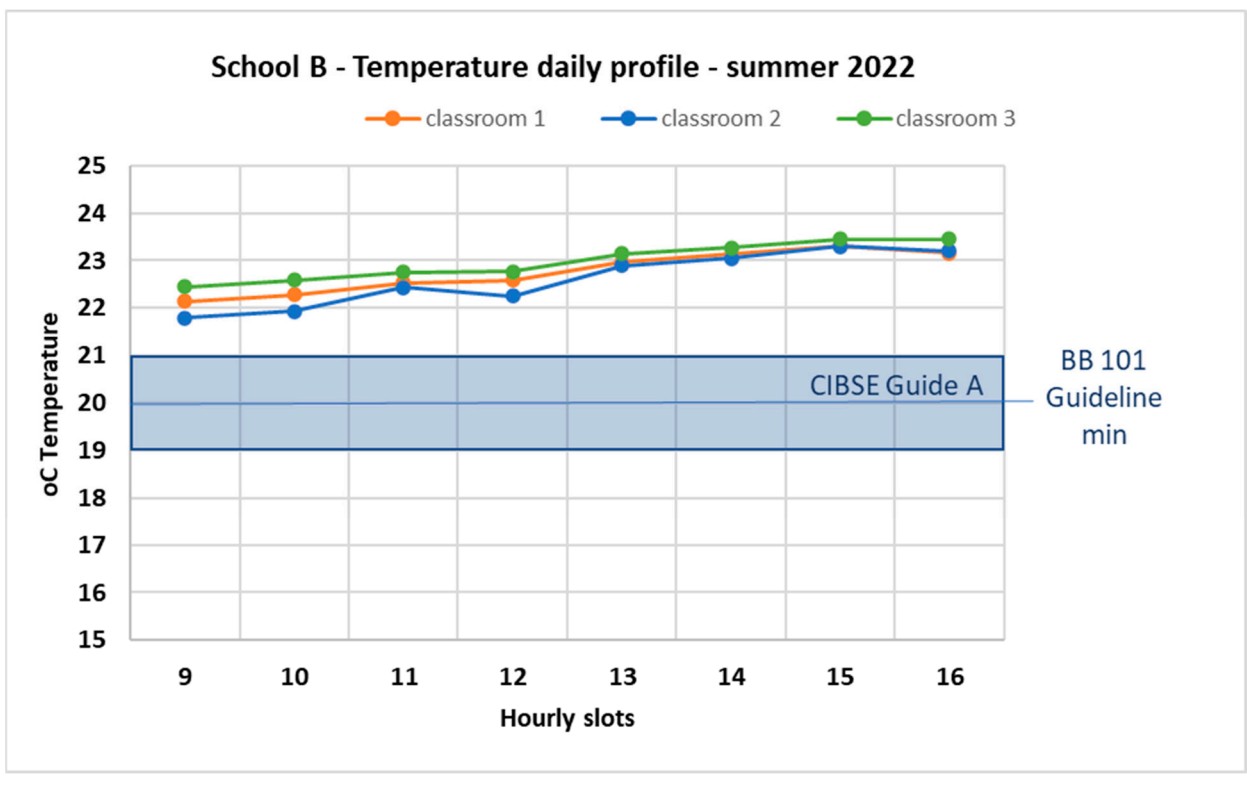

**Figure 12.** School B average temperature, hourly daily profile, summer.

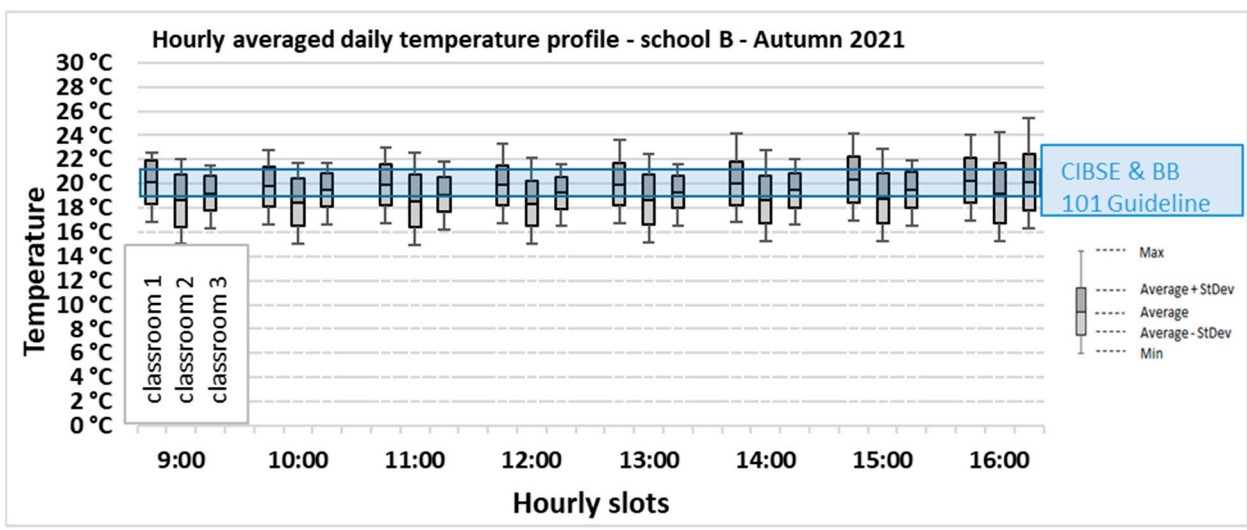

**Figure 13.** School B: temperature profile for 3 classrooms showing min, max, average temperatures per hour during autumnn of the academic year 2021/22.

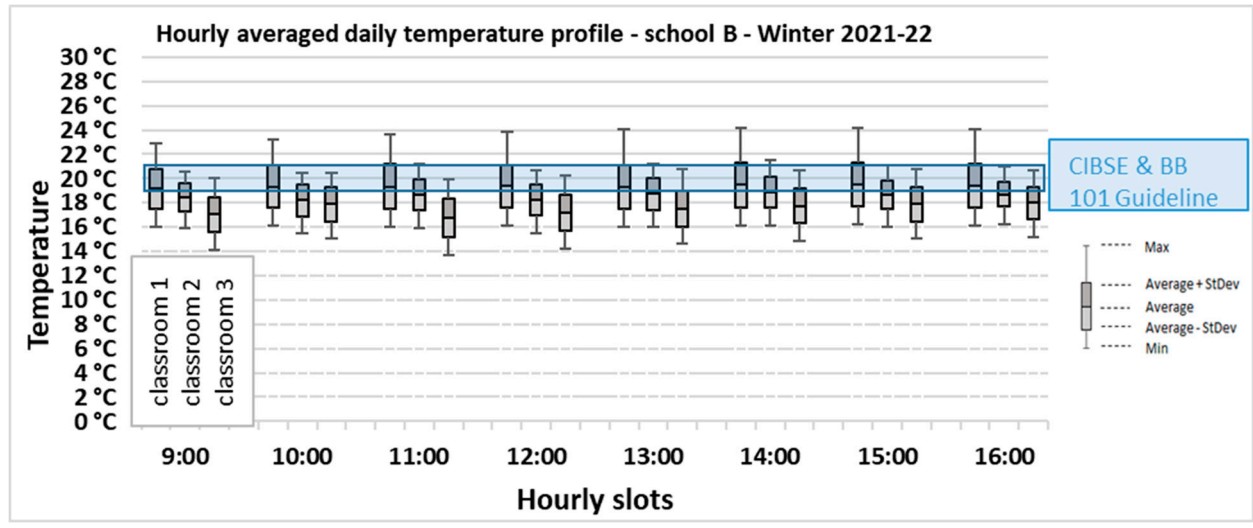

**Figure 14.** School B: temperature profile for 3 classrooms showing min, max, average temperatures per hour during winter of the academic year 2021/22.

*3.2. User Studies—Actions and Adaptations in Response to COVID-19 and Seasonal Comfort Perception—Teachers' Perspectives*

The teachers expressed their willingness to increase the ventilation of their classrooms in response to COVID-19. Ventilation was fostered by increasing the number of hours that windows and doors connected to outdoors were opened throughout occupied hours. Teachers recognised that this strategy was problematic for thermal comfort in winter as it resulted in cold classrooms. In order to balance the need to increase ventilation and maintain warm classrooms, windows and doors were reported to be open during times when pupils were not in the classroom; for example, before the start of the day, during breaks and during outdoor learning time. Teachers reported that they used the $CO_2$ monitors provided by Welsh Government to identify the $CO_2$ levels and schedule the time to open and close windows and external doors if $CO_2$ concentration levels rose. Teachers said that new layouts have been adopted in the classrooms where they primarily teach; for example, increasing the distance between desks and working stations, zoning learning/teaching activities inside the classrooms and redistributing furniture in the classroom. Teachers engaged with younger learners reported that learning activities prioritised more time for

outdoor learning, using covered areas connecting classrooms and outdoor open spaces. However, there was less flexibility in the use of communal spaces or teaching spaces outside the classrooms for example, the music room, the IT room and other learning spaces shared by different year groups. The strict timetable to use shared learning spaces was intended to limit the potential contact between different year groups. Teachers reported that advice had been provided to mitigate COVID-19 transmission risk. This advice included recommendations related to general school health and safety, cleaning, personal hygiene and social distancing. Teachers also reported other strategies adopted in response to COVID-19 such as staggered drop-off and collecting times, designated school access for different year groups and one-way circulation systems.

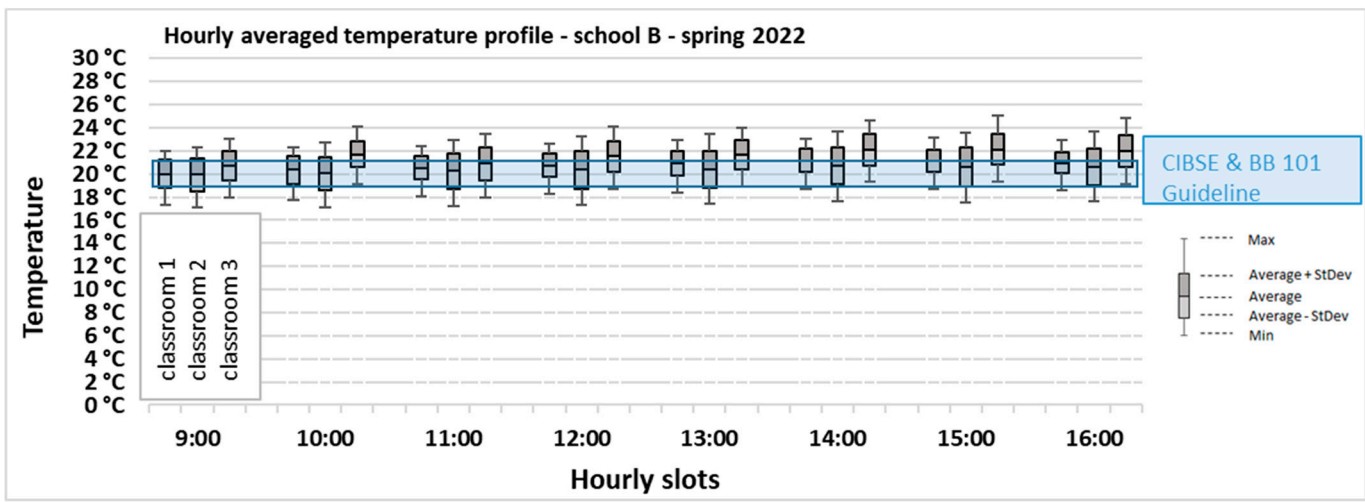

**Figure 15.** School B: temperature profile for 3 classrooms showing min, max, average temperatures per hour during spring of the academic year 2021/22.

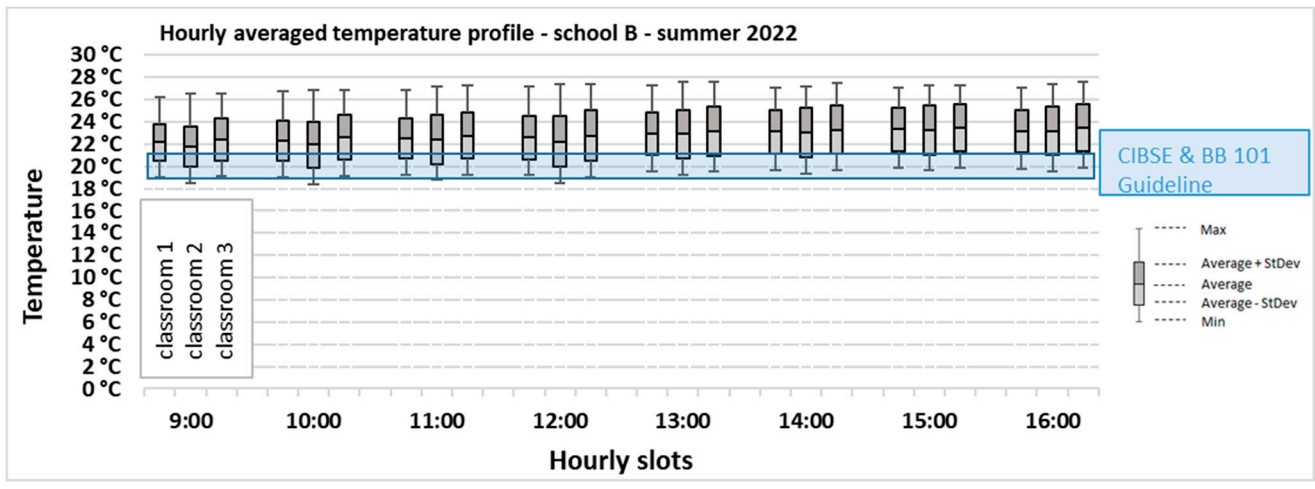

**Figure 16.** School B: temperature profile for 3 classrooms showing min, max, average temperatures per hour during summer of the academic year 2021/22.

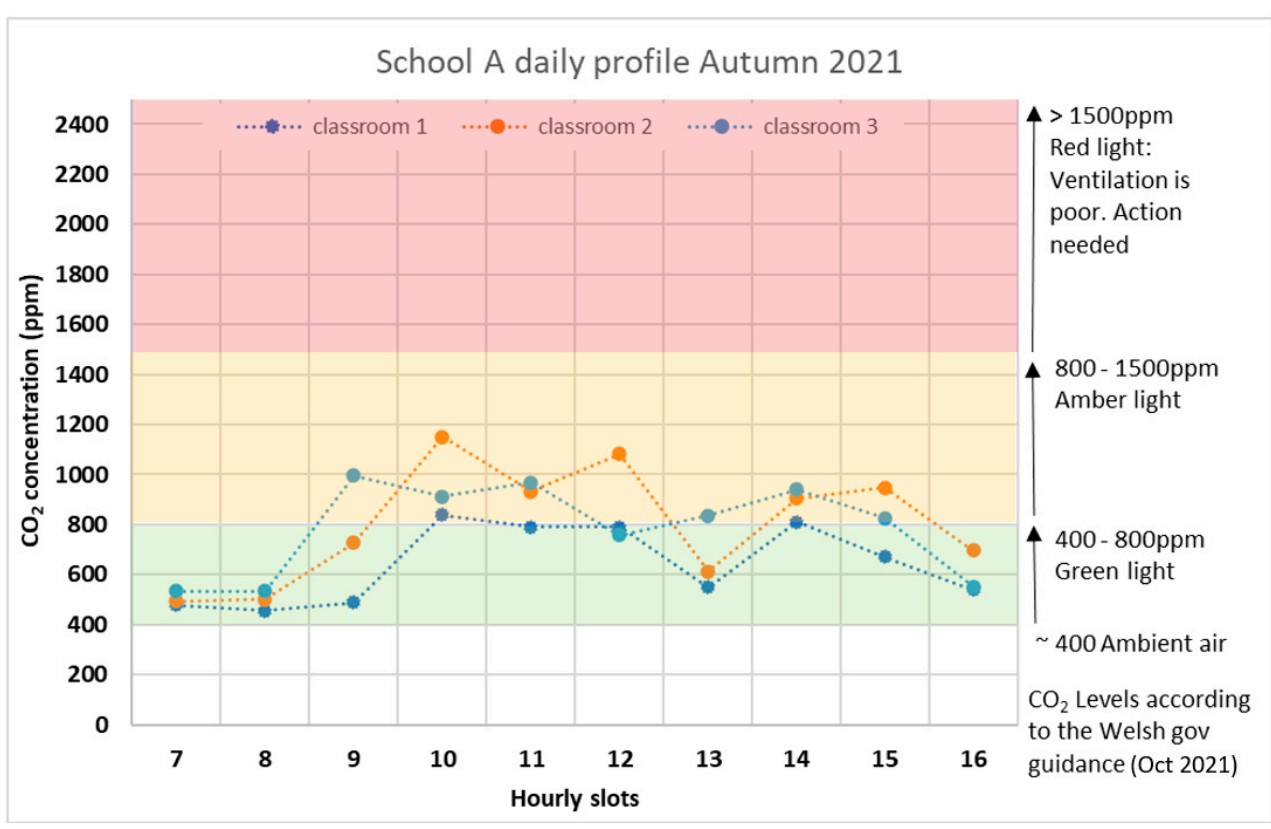

**Figure 17.** School A average $CO_2$ concentration (ppm) per classroom in autumn (weekdays, 7 a.m.–4 p.m.) [26].

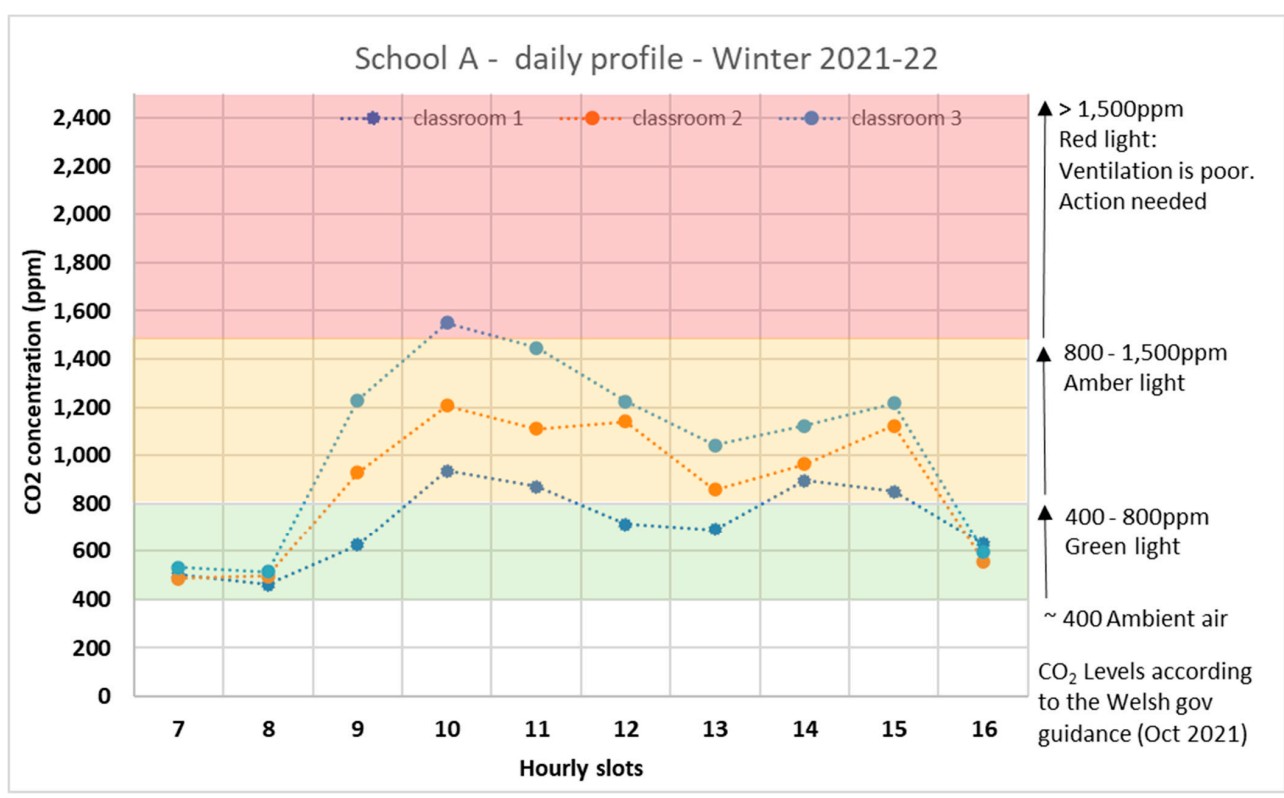

**Figure 18.** School A average $CO_2$ concentration (ppm) per classroom in winter (weekdays, 7 a.m.–4 p.m.) [26].

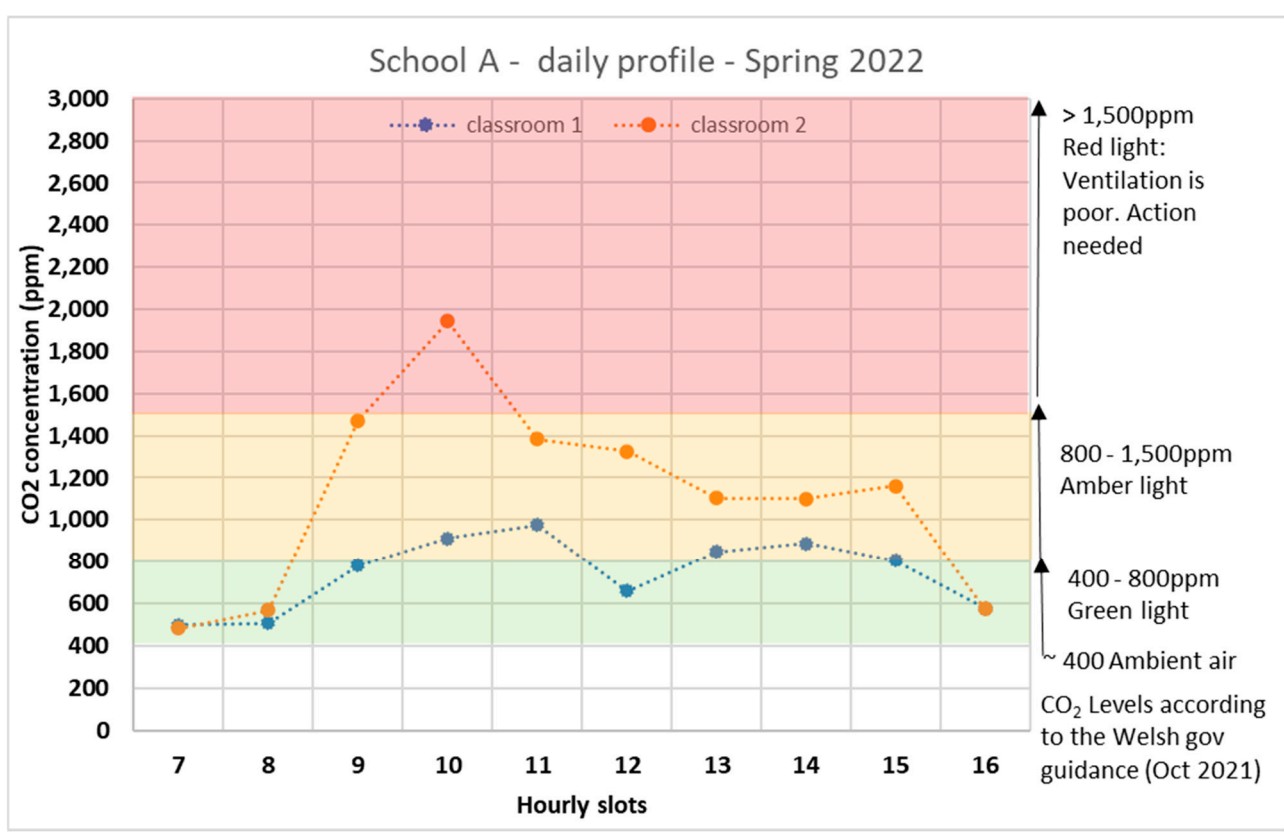

**Figure 19.** School A average CO$_2$ concentration (ppm) per classroom in the spring of 2022 (weekdays, 7 a.m.–4 p.m.) [26].

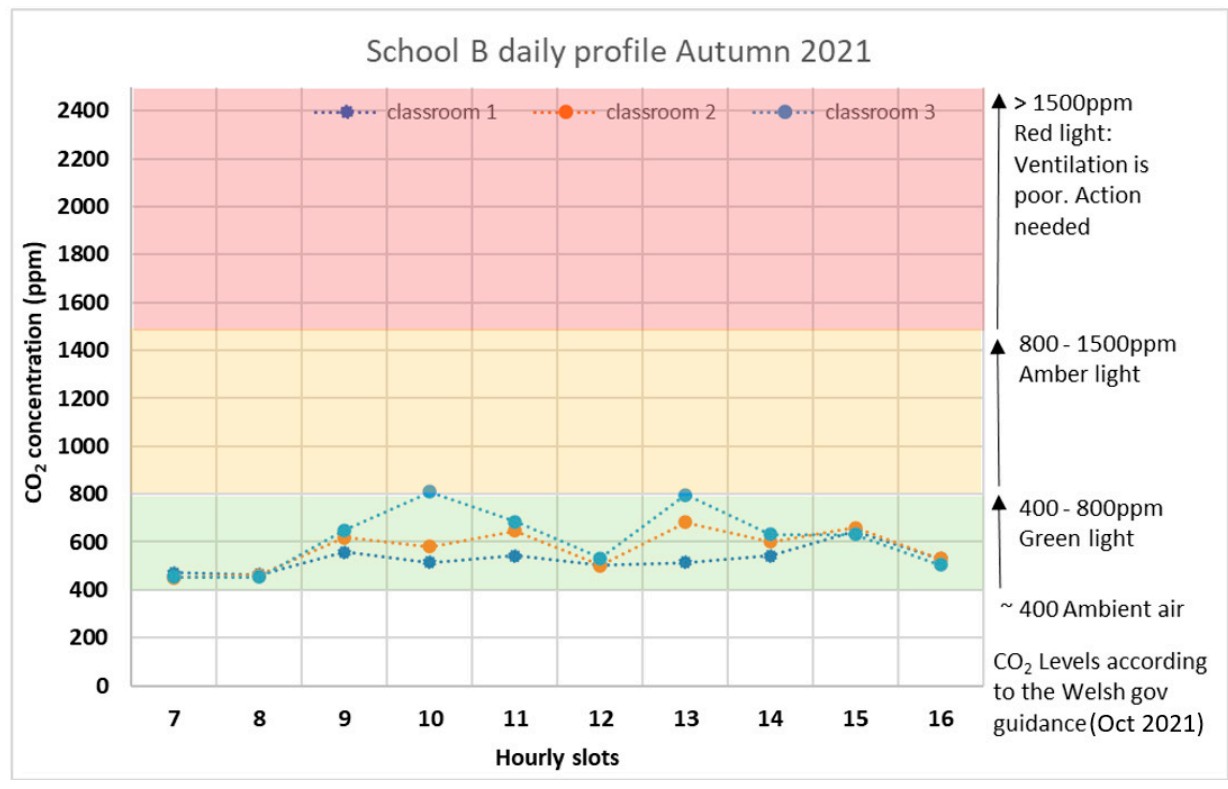

**Figure 20.** School B average CO$_2$ concentration (ppm) per classroom in autumn (weekdays, 7 a.m.–4 p.m.) [26].

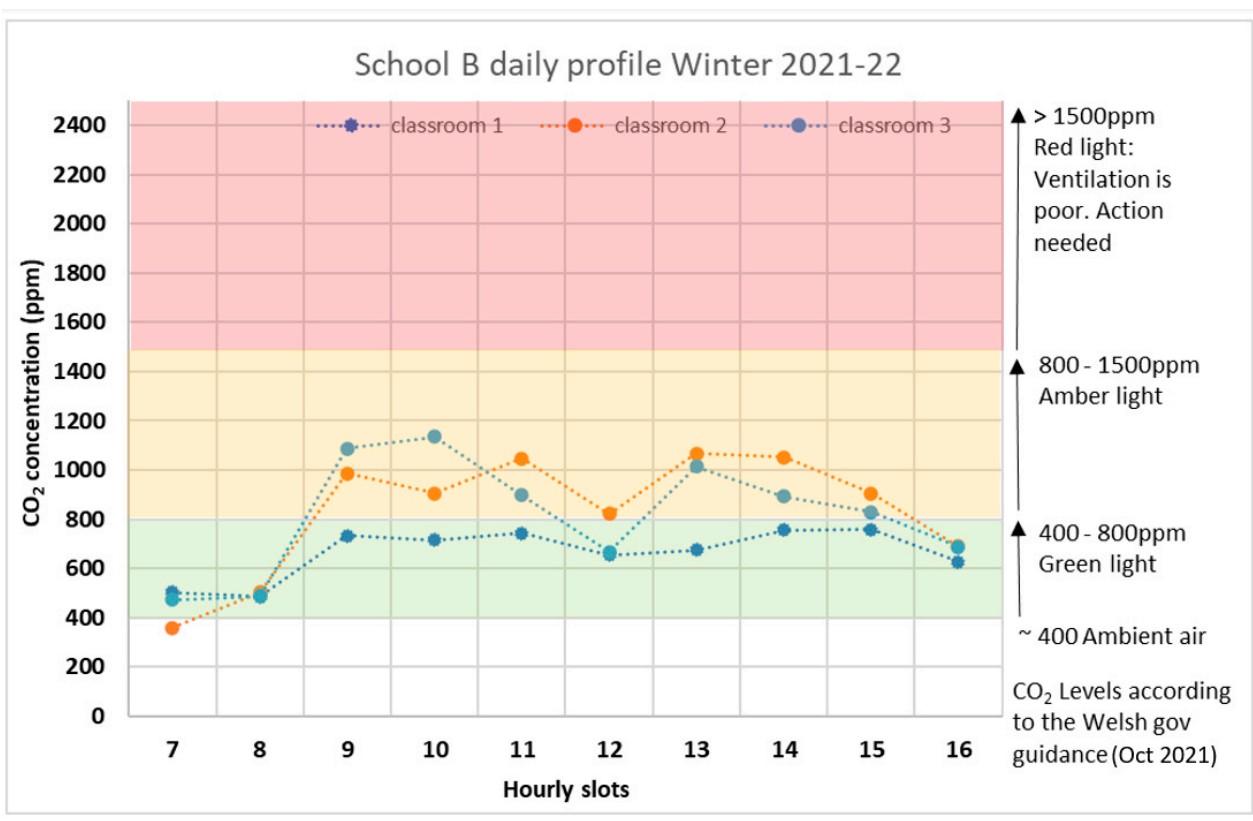

**Figure 21.** School B average $CO_2$ concentration (ppm) per classroom in winter (weekdays, 7 a.m.–4 p.m.) [26].

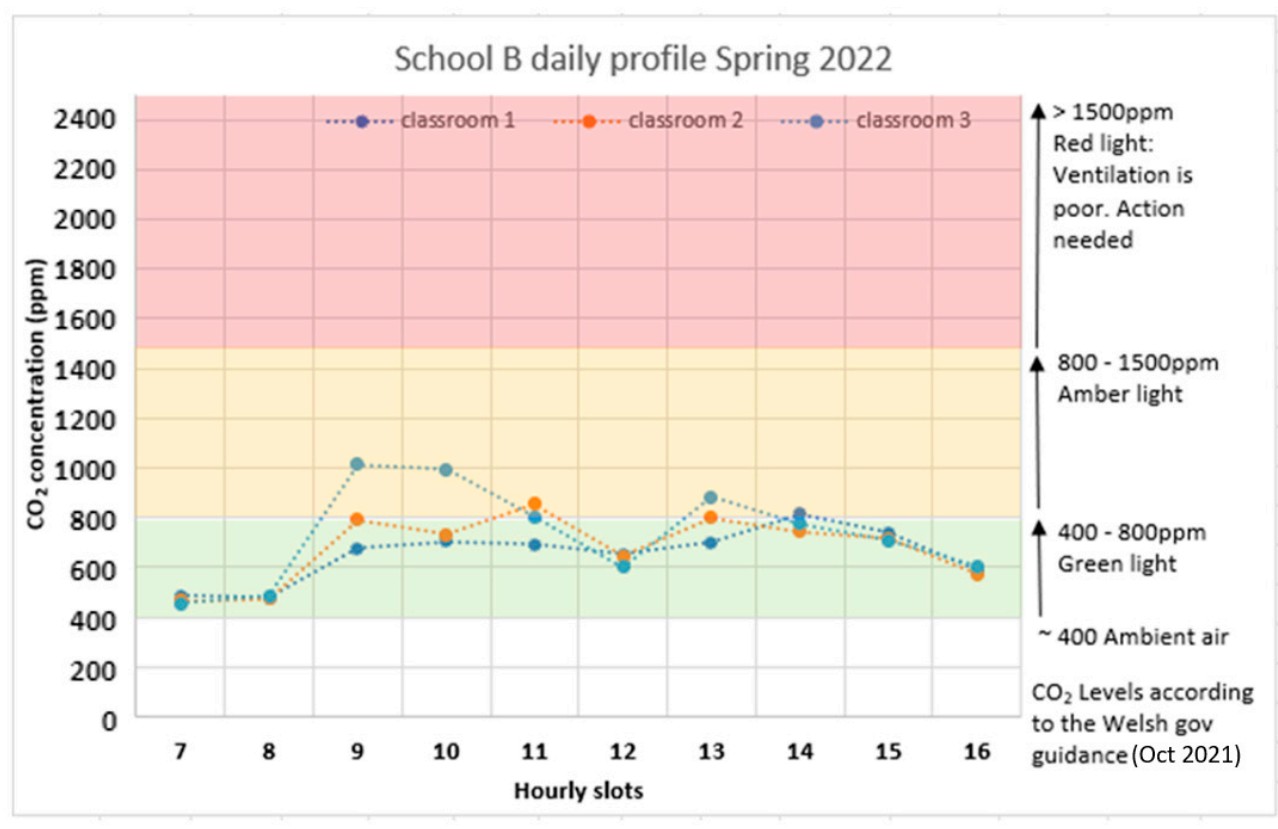

**Figure 22.** School B average $CO_2$ concentration (ppm) per classroom in the spring of 2022 (weekdays, 7 a.m.–4 p.m.) [26].

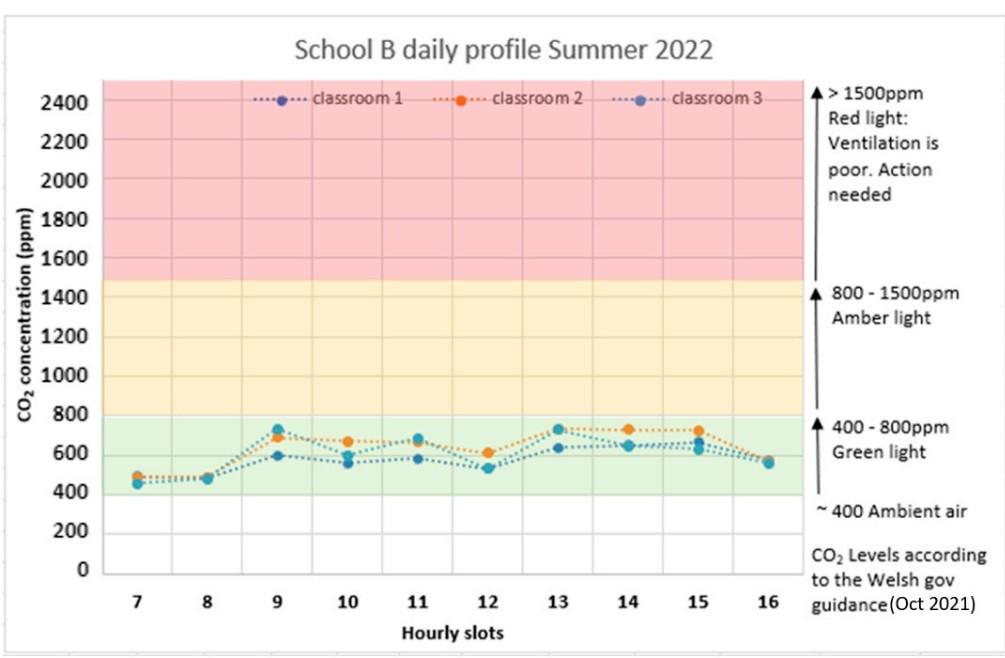

**Figure 23.** School B average $CO_2$ concentration (ppm) per classroom in the summer of 2022 (weekdays, 7 a.m.–4 p.m.) [26].

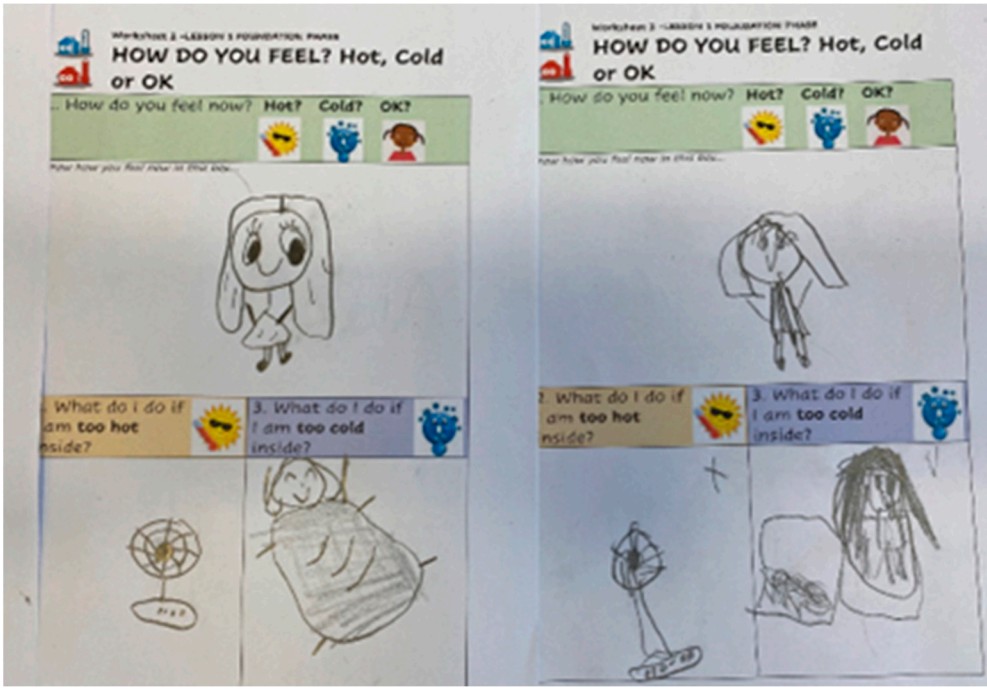

**Figure 24.** Pupils' drawings expressing their thermal perception by younger students 6–8 years old.

In terms of seasonal comfort perceptions reported by teachers in School A, 53% of respondents expressed that the temperature of their classroom was cold and that they felt slightly uncomfortable in winter. In terms of air freshness in winter, 46% of respondents felt that the air in the classroom was a bit fresh while 27% thought it was neither fresh nor unfresh. In School B, 50% of respondents expressed feeling slightly uncomfortable in winter because their classrooms were cold. 67% of respondents rated the air of their classrooms as being neither fresh nor unfresh. Summer responses in School A were spread across the comfort scale with 20% neither comfortable nor uncomfortable and 47% rating overall conditions in their classrooms in summer to be neither satisfactory nor unsatisfactory.

Summer responses in School B were also spread across the comfort scale with 50% of respondents expressing their classrooms felt hot in summer. The teachers in both case studies also rated the conditions in the milder seasons (autumn and spring). The teachers in School A expressed that the temperature of their classrooms felt neither comfortable nor uncomfortable (73%) and the air neither fresh nor unfresh (60%), while the teachers in School B expressed that the temperature of their classrooms felt neither comfortable nor uncomfortable (83%) and the air neither fresh nor unfresh (67%).

### 3.3. User Studies—Seasonal Comfort Perceptions—Pupils' Perspectives

The research explored the satisfaction levels of children and the actions taken to promote comfort via drawings and questionnaires. Table 3 summarised the monitored $CO_2$ levels and temperature on the day comfort studies were undertaken with the pupils. Due to health and safety guidance, the winter workshop in School B was not undertaken in the classroom where students are based. The children, however, were asked to reflect about the conditions of their classrooms during the workshops and the monitoring data illustrated in Table 3 correspond to classroom conditions, not to the condition of the space where the winter workshops in School B took place.

**Table 3.** Monitored $CO_2$ levels and temperature on the day where comfort studies were undertaken with pupils.

| Location | Winter | | Summer | |
|---|---|---|---|---|
| | Temperature °C Min–Max; Average | CO$_2$ ppm Min–Max; Average | Temperature °C Min–Max; Average | CO$_2$ ppm Min–Max; Average |
| School A | | | | |
| Classroom 1 | 14.9–16.6; 15.3 | 525–1200; 845 | 23.8–27.1; 25.1 | 550–1000; 625 |
| Classroom 2 | 15.3–17.9; 16.7 | 475–1350; 750 | 22.1–25.3; 24.1 | 500–2500; 881.25 |
| Classroom 3 | 14.9–16.5; 15.6 | 500–900 * | 22.1–24.5; 23.7 | 500–3000; 1221.43 |
| School B | | | | |
| Classroom 1 | 16.5–17.5; 16.85 ** | 500–900; 628.57 ** | 22.1–24.0; 23.09 | 475–780; 611,43 |
| Classroom 2 | 18.4–19.9; 19.04 ** | no data | 21.5–24.9; 23.23 | 550–970; 681.43 |
| Classroom 3 | 16.2–19; 18.10 ** | 650–1600; 1121.43 ** | 22.0–24.8; 23.50 | no data |

(*) Afternoon data only; (**) data from classrooms where children are based; however, winter workshop took place in a different space due to health and safety concerns.

Children used drawings to express their individual satisfaction levels with the thermal conditions and air freshness in their classroom and to identify the actions they take to be comfortable (Figures 24 and 25). The young children aged 6 to 8 years old were capable to communicate their thermal experience through drawings and discussions. The research instruments prompted them to reflect about their thermal experiences 'right here, right now'. However, the drawings of young children tend to illustrate thermal experiences and concepts inside and outside their classrooms, including the actions they take to be comfortable in their most familiar environments: their school classroom and their homes. Young children were able to identify a variety of personal actions they can take to be comfortable including changing their clothing levels (extra layers when cold, lighter clothes when hot), drinking cold drinks, using radiators and fans, opening and closing windows, using blankets to keep warm, etc.

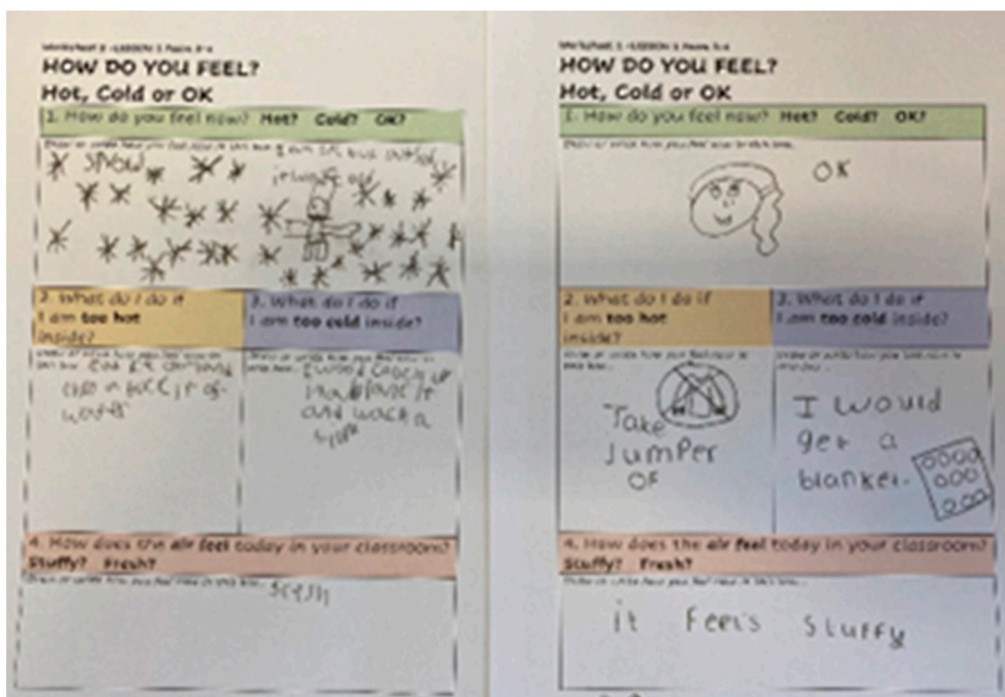

**Figure 25.** Pupils' drawings expressing their perceptions of indoor environment by students 9 years and older.

Older children (aged 9 years old and above) were also able to express through drawings their thermal experiences, their responses tend to focus on 'right here, right now' referring primarily to the satisfaction levels with the thermal conditions of their classroom. The actions that they identified were similar to those identified by younger children: changing their clothing levels, having cold and hot drinks and using radiators, fans and windows to control the temperature of their classrooms. In addition to the thermal experience, the older children also reflected about the freshness of the air and the ventilation adequacy of their classrooms. They were aware of the need to ventilate the classrooms and the main strategy they identified was the operation of windows and external doors to increase fresh air.

Comfort questionnaires administered to children 9 years and older explored their satisfaction levels with winter and summer conditions during the workshops. These responses were not disaggregated by gender as potential differences in gender-based responses are irrelevant for the context and aims of this study [35]. The comfort questionnaires are discussed in relation to the temperature and $CO_2$ levels monitored in the classrooms. The locations of sensors and monitoring equipment are illustrated in Figures 26–34.

In terms of thermal comfort in December, most of the pupils in School A felt comfortable (67% of respondents), 26% of pupils felt very cold and 7% felt very hot. In terms of air freshness, 71% of pupils considered that the air in the classroom was fresh, 7% felt that the air was unfresh and 22% felt that the air was neither fresh nor unfresh. Monitoring data collected in the classrooms on the survey day show that temperature in the classrooms evaluated by pupils range from about 15 °C to 17 °C and $CO_2$ levels are between 475–1350 ppm in winter.

The results of the survey in December in School B show that 34% of pupils felt very cold, 26% felt comfortable and 40% felt very hot. In terms of air freshness, 30% considered that the air in the classroom was fresh, 30% considered the air was unfresh and 40% thought that the air was neither fresh nor unfresh. No monitoring data are available for the room where the workshops took place.

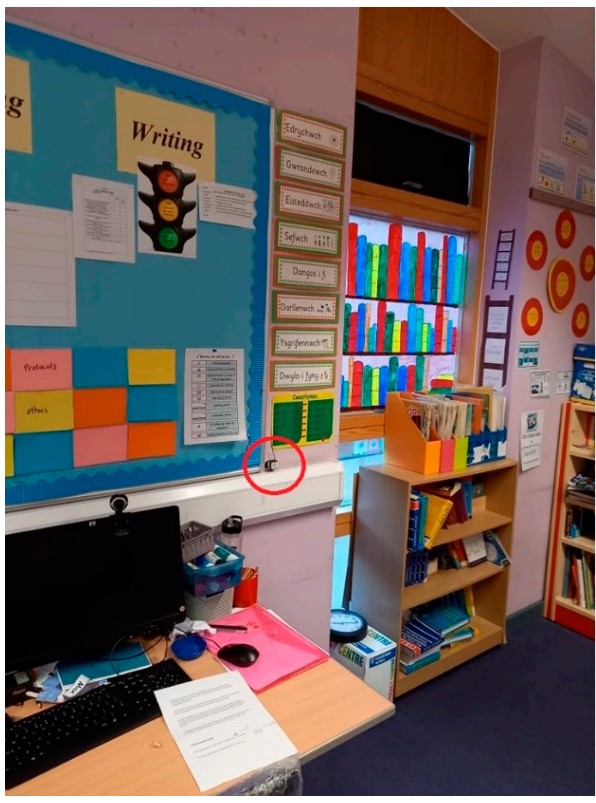

**Figure 26.** T/RH sensor in School B, Classroom 1.

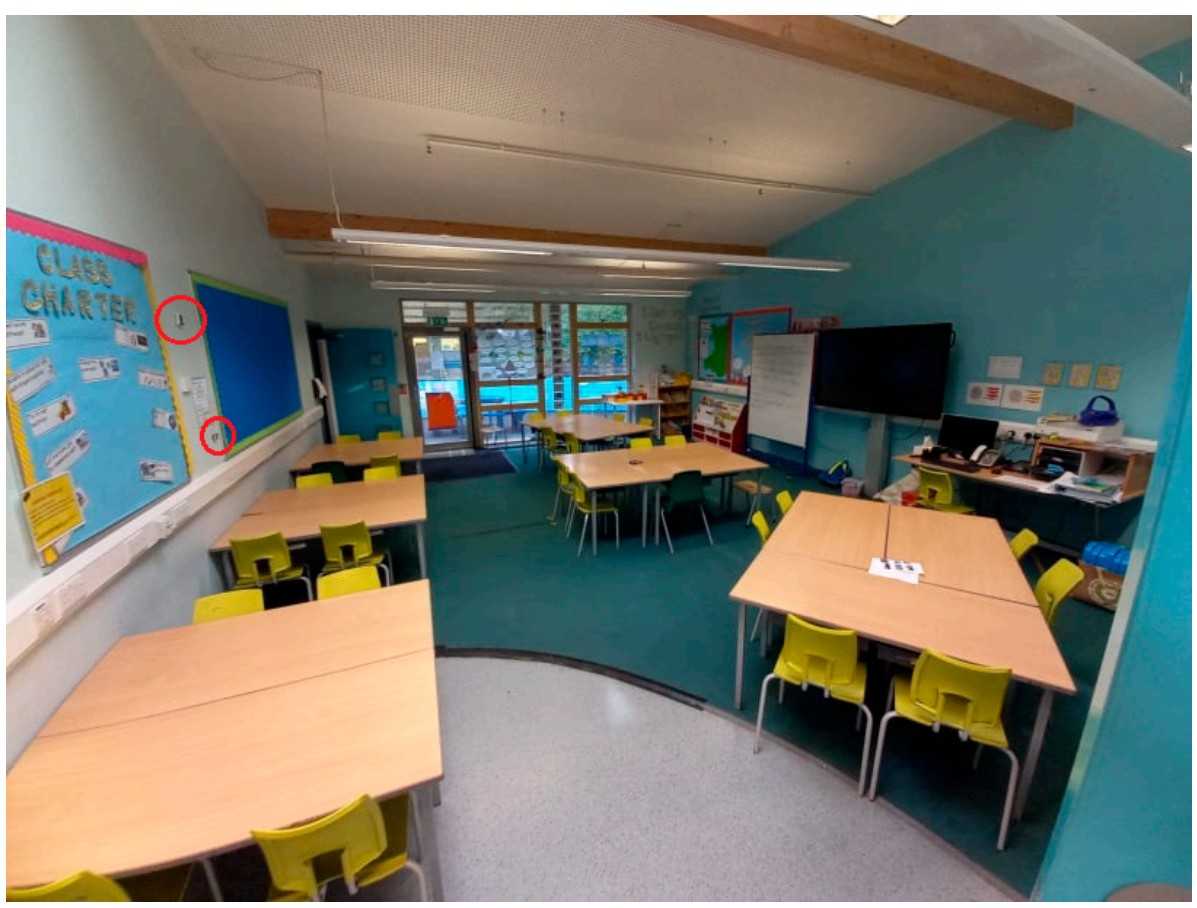

**Figure 27.** T/RH and CO$_2$ sensors in School B, Classroom 2.

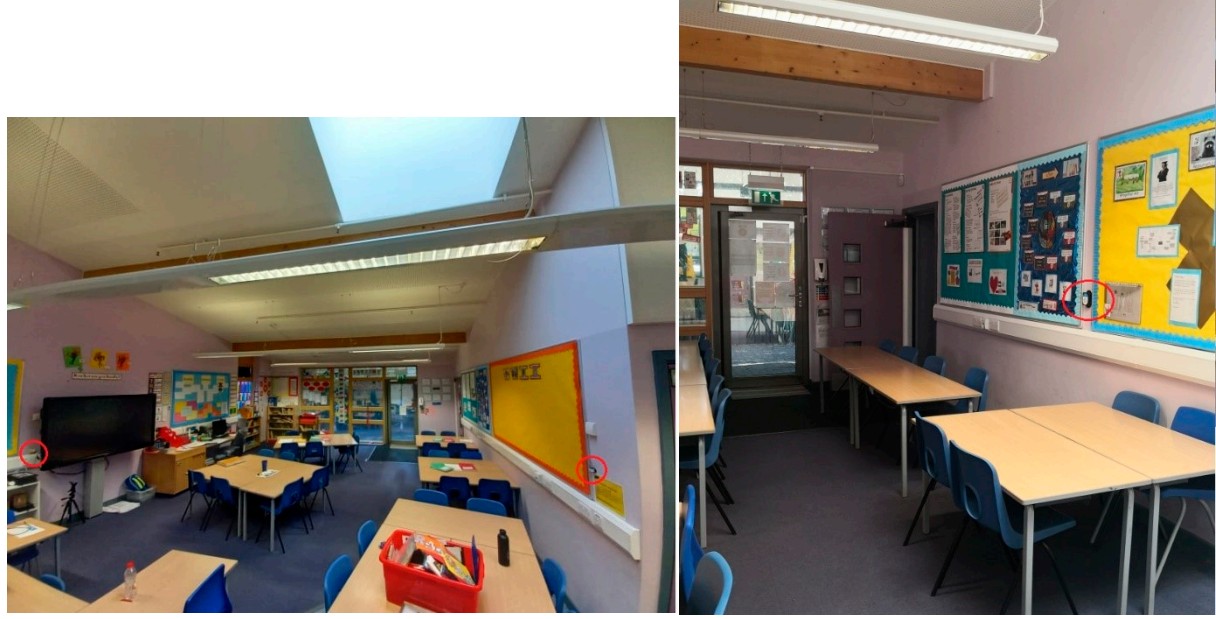

**Figure 28.** T/RH and CO$_2$ sensors in school B, Classroom 3.

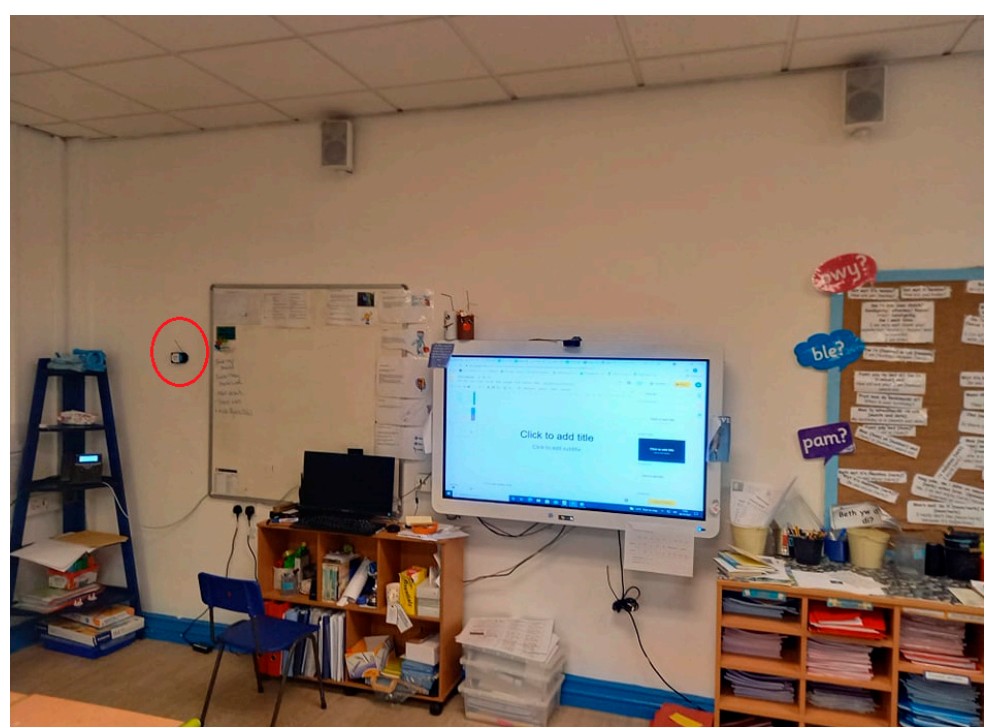

**Figure 29.** CO$_2$ sensor in School A, Classroom 3.

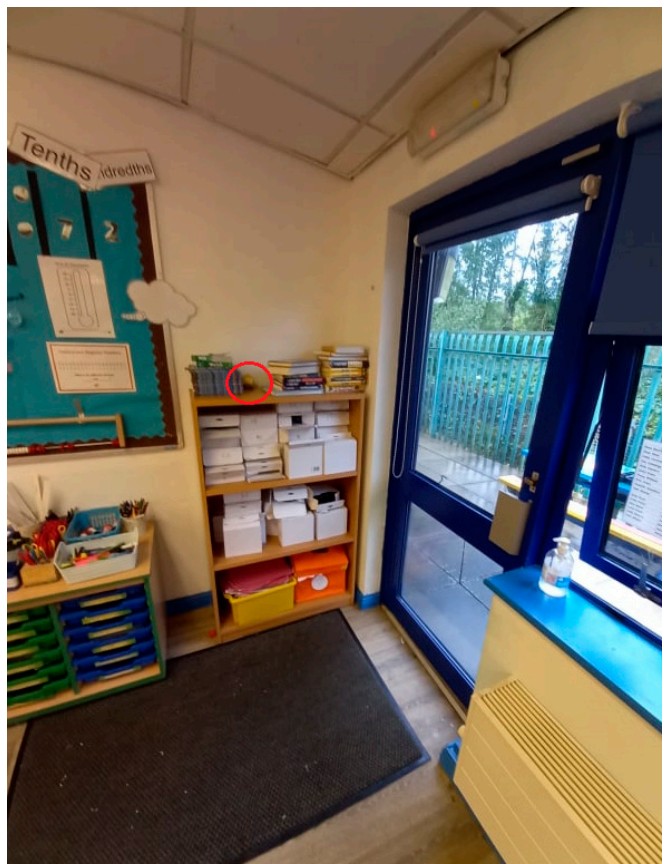

**Figure 30.** T/RH sensors in School A, Classroom 3.

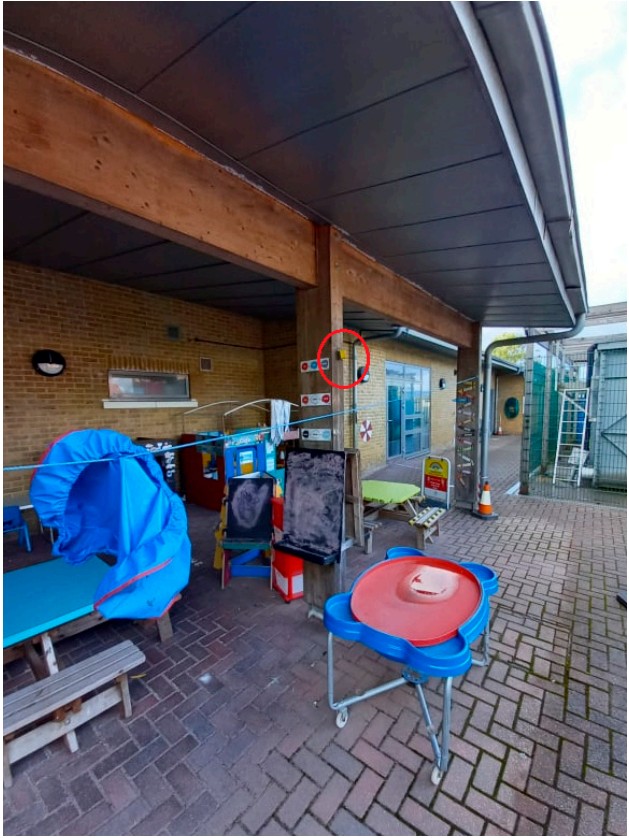

**Figure 31.** Outdoor T/RH sensors in School B.

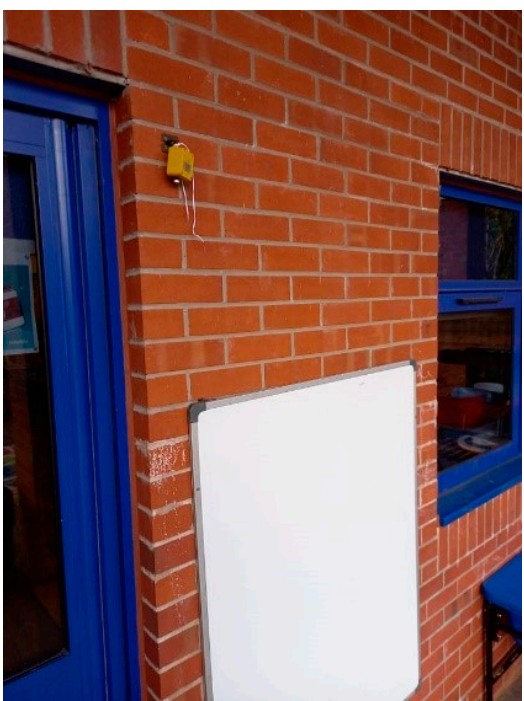

**Figure 32.** Outdoor T/RH sensors in School A.

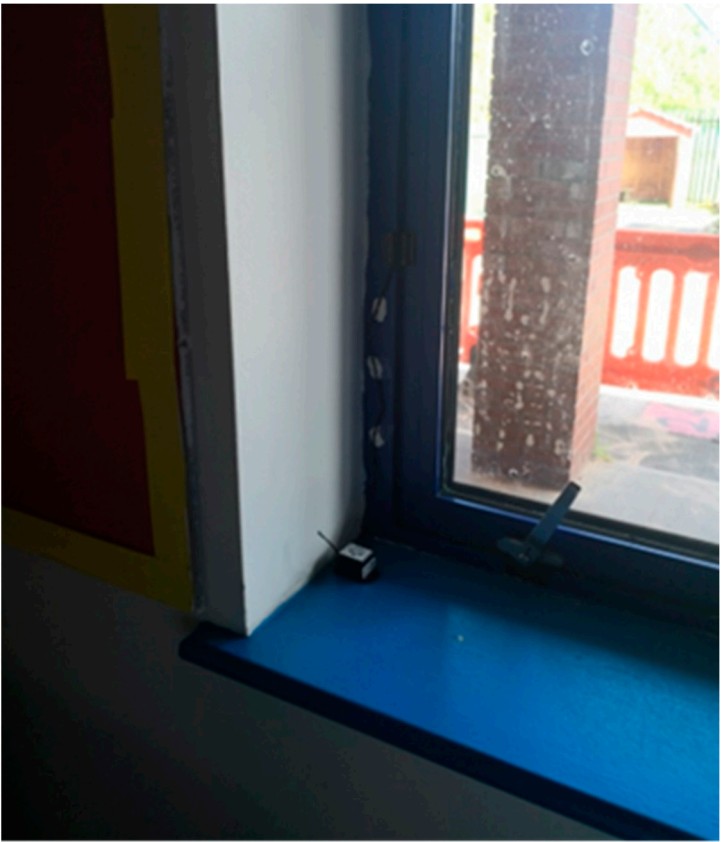

**Figure 33.** Window open/closed status installed sensor in School A.

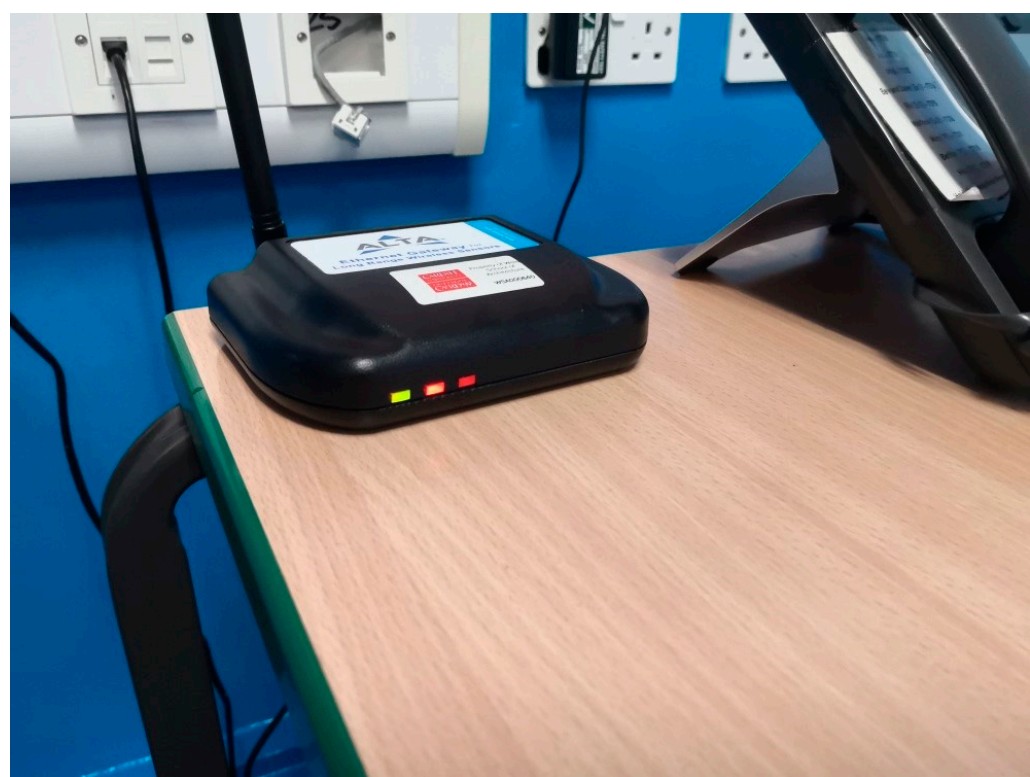

**Figure 34.** The gateway for receiving and transmitting the monitoring data.

The survey results in summer show that in School A, 50% of pupils felt comfortable with the temperature of their classroom and 42% felt very hot. In terms of air freshness, 48% thought the air was neither fresh nor unfresh, and 44% thought the air was fresh. Monitoring data collected in the classrooms on the survey day show that temperature in the classrooms evaluated by pupils range from about 21.5 °C to 25 °C and $CO_2$ levels are between 500–3000 ppm in summer. In School B, 75% of pupils felt very hot in their classroom and 23% felt comfortable. In terms of air freshness, 52% pupils reported that the air in the classrooms felt fresh and 45% thought the air in their classroom was neither fresh nor unfresh. Monitoring data collected in the classrooms on the survey day show that temperature in the classrooms evaluated by pupils range from about 22 °C to 27 °C and $CO_2$ levels are between 475–970 ppm in summer.

## 4. Discussion and Conclusions

Schools have had to take measures and to implement health and safety guidance and strategies to respond to COVID-19 in order to ensure that children learn in safe and healthy school environments. In the first academic year after schools returned to teaching in school buildings, the predominant strategies to mitigate COVID-19 risks relied on adaptations and changes in the way school spaces were used, in patterns and routines of operation and control (for example operation of windows and doors to increase ventilation). Some of these strategies, such as the use of windows to promote fresh air in classrooms in the heating season can, however, be detrimental to the thermal comfort in classrooms and potentially lead to increased energy use for heating.

Overall, the monitoring data suggest that the indoor conditions of the classrooms in terms of temperature and $CO_2$ concentration levels were acceptable most of the year, and did not prompt serious concerns about the overall indoor quality. $CO_2$ concentration far above the accepted range was recorded during winter and spring, due to less natural ventilation. While a high $CO_2$ concentration depicts inadequate ventilation, it is not uncommon and has been recorded in similar studies (e.g., [36]) In terms of thermal comfort, discomfort was reported in winter due to classrooms being cold. In the winter season,

monitoring data show that classrooms temperature were slightly lower than CIBSE Guide A and Building Bulletin 101. Long term monitoring data recorded in spring in School A show that temperatures were significantly lower than CIBSE Guide A and Building Bulletin 101 recommendations. The pupils' responses generally show that 67% of students felt comfortable in School A (average temperatures between 16 °C and 18 °C) while in School B, pupils' responses showed that only 26% of pupils reported to feel comfortable with 40% of pupils feeling very hot and 34% very cold. Indoor temperatures in classrooms was measured in the range of 17 °C and 19.5 °C in winter at the time of the surveys. Summer temperatures range from 21 to 27 °C on the day of the summer surveys, with the majority of students expressing that they felt too warm in both schools under the summer conditions recorded on the day of the survey. Relative humidity was within the acceptable range of 40–60% CIBSE Guide A for most times during occupied period. Previous research has shown that too high and too low relative humidity affects buildings occupants' perception of indoor temperature which in turn affects their thermal comfort [37]. In addition, relative humidity outside of this recommendation can encourage perceived poor air quality, including the emergence of 'sick building syndrome' such as itchy skin and eye irritation [38].

The $CO_2$ levels remained between 500 and 1500 ppm during the majority of occupied hours in the monitored classrooms. The profile and trends of $CO_2$ concentration levels in the classrooms monitored in this study relate to the findings of the study by Monge-Barrio et al. [39] where $CO_2$ levels were measured in classrooms in Northern Spain post COVID-19 in the heating season. The monitored data in this study points out at instances where $CO_2$ levels exceed 1500 ppm at different times of the day during occupied hours. However, these results are expected given the size of the classrooms, their density and the main ventilation strategy deployed, reliant on opening windows and external doors. It is worth noting that previous research measuring $CO_2$ levels in schools has found that $CO_2$ concentration levels in classrooms tend to exceed 1000 ppm [3,40]. When exploring the seasonal differences in profile of $CO_2$ concentration levels, the monitoring data suggest that $CO_2$ levels in School A tend to be lower in in autumn compared to other seasons, with the majority of monitoring data showing that $CO_2$ levels tend to remain below 1500 ppm (green and amber zones by Welsh Government guidance). In School B, the monitoring data suggest that $CO_2$ levels tend to remain below 800 ppm in all classrooms in autumn, spring and summer. Research undertaken by Deng and Lau [41] pre COVID-19 surveyed 220 classrooms in Midwest USA and found that $CO_2$ levels were below 1000 ppm only in 96 classrooms in autumn (44% of classrooms in the study) and 70 classrooms in winter (32% of classrooms in the study). Exposures to $CO_2$ levels above 1000 pm were found to be typical in the surveyed schools in the study by Deng and Lau [41], aligned to monitoring results in this work.

The profile of $CO_2$ concentration levels is indicative of actions related to classroom activities and those reported by teachers: $CO_2$ levels rise after the start of the day due to occupation; $CO_2$ levels decrease during lunch breaks and at the end of the school day. However, in general $CO_2$ concentration levels in most of the monitored classrooms tend to remain below 1200 ppm, suggesting that ventilation levels in classrooms were generally acceptable with minor interventions such as window opening adopted on a regular basis to increase the freshness of the air. However, it should be noted that the natural ventilation strategy to promote ventilation can be problematic in relation to thermal comfort and may result in extremely low temperatures as shown by the long-term monitoring data in spring in School A. The profile of $CO_2$ concentration levels in classrooms were predominantly within the green and amber light zones as per Welsh Government guidelines in autumn and in winter, showing that simple actions adopted by teachers helped to manage the $CO_2$ concentration levels indoors in the monitored classrooms during that period. Data from School A show that a number of occupied hours in classrooms during spring were predominantly within the amber and red-light zones suggesting that actions were ineffective to manage and reduce the $CO_2$ concentrations and lead to thermally

uncomfortable spaces in the coldest season. These classrooms were located in schools built in the last 10–20 years with a DEC rating of D (typical energy performance for the building type) and graded A in terms of condition and suitability rating.

Whilst the operation of windows and external doors promoted fresh air and ventilation in the classrooms to $CO_2$ levels commensurate to green and amber zones, this strategy required teachers to consider the thermal comfort conditions in different seasons to balance the need to keep warm indoors and decrease $CO_2$ concentration levels. During winter and spring seasons, when outdoor temperatures dropped, windows were predominantly open when pupils were not in the classroom to avoid discomfort due to cold temperature in the classroom. Windows and external doors were open before the start of the school day, during lunch breaks and outdoor play times and were kept open after the end of the school day. This seemingly led to an increase in $CO_2$ concentration levels in winter as compared to other seasons.

This study shows that teachers and pupils, as school end-users, were interested in promoting good ventilation and engaged with strategies to foster fresh air indoors and thermal comfort indoors. The work indicates that end-users were willing to adopt changes and actions to exert personal adaptations to achieve thermal comfort, promote fresh air by interacting with building technologies, informed by COVID-19 guidance. These insights question the notion that end-users are uninterested and unable to take action to manage indoor environments to achieve performance goals ($CO_2$ concentration levels and thermal comfort). In these case studies, teachers reported the use of $CO_2$ monitors in their classroom and considered their own and their pupils' experiences of (dis)comfort to inform personal adaptation, use of controls and building features to manage the indoor environment. Pupils as young as 6 years old were able to express their comfort and discomfort and identify personal strategies to manage their thermal experience. They were also adept at finding ways that building technologies and architecture design features may affect their thermal environment and the freshness of the air in their classrooms. Children were naturally curious to learn, experiment and explore the implications of different $CO_2$ concentration levels and thermal conditions and identify actions to maintain acceptable indoor conditions in their classrooms, and some of the actions highlighted awareness of their personal agency, while other strategies relied on adults to take action on behalf of the children.

This work has undertaken a monitoring study and user studies to explore seasonal variations in the $CO_2$ concentration levels and thermal conditions in a small number of classrooms in two primary schools in South Wales. The monitoring data have been analysed in relation to user studies that explore the pupils' and teachers' satisfaction levels with the indoor conditions in winter and in summer and identified actions taken to promote comfort and ventilation in response to COVID-19 concerns. The work has identified that teachers and schools have adopted a number of actions and responses to minimise COVID-19 transmission risk indoors which included modifications in the layout of classrooms, use of spaces, monitoring of $CO_2$, following COVID-19 health and safety guidance and increased operation of windows and doors to promote ventilation. The monitoring data show that overall, classrooms were able to maintain acceptable $CO_2$ concentration levels and tolerable thermal conditions for the majority of the occupied hours with some difficulties in ventilating and keeping warm indoors in the winter and spring seasons. The education and guidance surrounding COVID-19 prevention and the provision of feedback via $CO_2$ monitors was perceived as helpful to prompt actions to manage $CO_2$ concentration level indoors. This simple but effective behavioural intervention has tapped onto the end-users' willingness and ability to manage their indoor environments to promote healthy comfortable spaces in response to COVID-19 pandemic. It highlights opportunities for learning, education and engagement with building end-users and occupants to promote behavioural interventions to manage indoor environments and building performance and to promote actions of different users, such as pupils and teachers, to achieve comfortable and healthy indoor environments. Approaches that consider the end-user as an active participant of the interventions, with potential agency and motivations to take action as

a response to societal concerns such as health and wellbeing in the face of COVID-19 threats, or energy efficiency and energy use reduction in the face of climate emergency, may take significant steps towards better performing and sustainable buildings, particularly if end-users are empowered and supported in adopting positive actions. The combination of monitoring data to provide feedback and facilitate awareness and education together with the reflection of opportunities afforded by personal action and simple no-cost or low-cost technology use may offer effective first steps to address the impact of buildings and built environment on climate change.

This work conducted research with a small number of classrooms, hence the indoor environmental conditions of school buildings remain largely unknown, particularly if considering the high variety of quality, conditions and architectural features of educational buildings, such as primary schools. Hence, more research is required to identify building indoor environmental conditions and performance and also to explore no-cost and low-cost behavioural interventions together with more ambitious retrofitting and maintenance interventions to improve indoor conditions and promote healthy classrooms in energy efficient ways, especially in situations when simple actions by end-users do not suffice to achieve acceptable conditions and energy efficiency to levels required by climate emergency, and by the demands of net zero carbon aspirations in the built environment.

This was a relatively small-scale study and therefore a significant limitation of the data is the number of schools that could be feasibly monitored, and therefore the diversity of school buildings of different architectural styles and design features in Wales and the UK more widely are not represented. A relatively small number of children and teachers were included in the surveys. It was not possible to record with high precision the timing of the school activities and the number of pupils at each activity. Although there is a generic daily schedule for each class, there are activities in different rooms of the schools and the number of the pupils vary throughout the school year due to sick absences or splitting groups for different activities. The environmental parameters ($CO_2$ concentration, temperature and relative humidity) are affected by occupancy. Future work will investigate the architectural characteristics of the classrooms that affect the $CO_2$ concentrations levels and the thermal environment, integrating monitoring data and perceived satisfaction levels reported by pupils and teachers. It will engage with young children and explore the use of monitoring data to raise awareness, experiment and explore the connections between actions taken in spaces we use in everyday life, end-user's agency and resulting building performance with the aim to articulate an acceptable balance between health, comfort, energy efficiency and, more broadly, sustainability in school buildings.

**Author Contributions:** Conceptualisation, M.G.Z.-L.; data curation, M.I., O.T. and M.G.Z.-L.; formal analysis, M.G.Z.-L., M.I. and O.T.; funding acquisition, M.G.Z.-L. and T.A.S.; investigation, M.G.Z.-L., M.I., O.T. and T.A.S.; methodology, M.G.Z.-L., T.A.S. and M.I.; project administration, M.G.Z.-L.; resources, M.G.Z.-L. and T.A.S.; supervision, M.G.Z.-L. and T.A.S.; visualisation, M.I., O.T. and M.G.Z.-L.; writing—original draft, M.G.Z.-L.; revisions M.G.Z.-L., M.I., O.T. and T.A.S. All authors have read and agreed to the published version of the manuscript.

**Funding:** This research was funded by EPSRC Impact Accelerator Account grant number 521026 and ESRC Impact Accelerator Account grant number 522302.

**Institutional Review Board Statement:** The study was conducted in accordance with UKRI Research ethics guidance, Research Integrity and Ethical principles outlined by Cardiff University, and approved by the Ethics Committee of the Welsh School of Architecture, Cardiff University (SREC No 2142 on 8 October 2021).

**Informed Consent Statement:** Informed consent was obtained from all subjects involved in the study.

**Data Availability Statement:** Information on the data underpinning the results presented here, including how to access them, can be found in the Cardiff University data catalogue at http://doi.org/10.17035/d.2023.0246886931. This paper expands the research initially reported in Zapata-Lancaster and Ioanis [42]. This publication has significantly elaborated and expanded the scope, aims and analysis presented in Zapata-Lancaster and Ioanis [42], leading to an original research contribution.

**Conflicts of Interest:** The authors declare no conflict of interest.

## Appendix A. Comfort in Schools: Teacher Survey

**Perceptions of indoor environment in teaching/learning spaces**

*Date & time* ________________________________________________

*Year group you teach* ___________________________________

**Your opinion about the conditions <u>in the classroom you primarily teach</u>**

1. <u>*How do you describe the typical conditions in the classroom you primarily teach in WINTER?*</u>

*Temperature in winter*

|  | *Please tick* |  |  |  |  |  |
|---|---|---|---|---|---|---|
| Uncomfortable | 1 | 2 | 3 | 4 | 5 | Comfortable |

|  | *Please tick* |  |  |  |  |  |
|---|---|---|---|---|---|---|
| Too hot | 1 | 2 | 3 | 4 | 5 | Too cold |

|  | *Please tick* |  |  |  |  |  |
|---|---|---|---|---|---|---|
| Stable | 1 | 2 | 3 | 4 | 5 | Varies during the day |

*Air in winter*

|  | *Please tick* |  |  |  |  |  |
|---|---|---|---|---|---|---|
| Still | 1 | 2 | 3 | 4 | 5 | Draughty |

|  | *Please tick* |  |  |  |  |  |
|---|---|---|---|---|---|---|
| Dry | 1 | 2 | 3 | 4 | 5 | Humid |

|  | *Please tick* |  |  |  |  |  |
|---|---|---|---|---|---|---|
| Fresh | 1 | 2 | 3 | 4 | 5 | Stuffy |

|  | *Please tick* |  |  |  |  |  |
|---|---|---|---|---|---|---|
| Odourless | 1 | 2 | 3 | 4 | 5 | Smelly |

*Overall conditions in winter*

|  | *Please tick* |  |  |  |  |  |
|---|---|---|---|---|---|---|
| Unsatisfactory overall | 1 | 2 | 3 | 4 | 5 | Satisfactory overall |

2. <u>*Please add any additional comments you have about heating, temperature, air and related indoor environment conditions in the classroom in winter*</u>

--------------------------------------------------------------------------------------------------------------------

--------------------------------------------------------------------------------------------------------------------

--------------------------------------------------------------------------------------------------------------------

3. <u>*How do you describe the typical conditions in the classroom you primarily teach in SUMMER?*</u>

<u>*Temperature in summer*</u>

|  | *Please tick* |  |  |  |  |  |
|---|---|---|---|---|---|---|
| Uncomfortable | 1 | 2 | 3 | 4 | 5 | Comfortable |

|  | *Please tick* |  |  |  |  |  |
|---|---|---|---|---|---|---|
| Too hot | 1 | 2 | 3 | 4 | 5 | Too cold |

|  | *Please tick* |  |  |  |  |  |
|---|---|---|---|---|---|---|
| Stable | 1 | 2 | 3 | 4 | 5 | Varies during the day |

*Air in summer*

*Please tick*

| Still | 1 | 2 | 3 | 4 | 5 | Draughty |
|---|---|---|---|---|---|---|

*Please tick*

| Dry | 1 | 2 | 3 | 4 | 5 | Humid |
|---|---|---|---|---|---|---|

*Please tick*

| Fresh | 1 | 2 | 3 | 4 | 5 | Stuffy |
|---|---|---|---|---|---|---|

*Please tick*

| Odourless | 1 | 2 | 3 | 4 | 5 | Smelly |
|---|---|---|---|---|---|---|

*Conditions in summer*

*Please tick*

| Unsatisfactory overall | 1 | 2 | 3 | 4 | 5 | Satisfactory overall |
|---|---|---|---|---|---|---|

4. <u>**Please add any additional comments you have about cooling, air, ventilation and related indoor environment conditions in the classroom in summer**</u>

-------------------------------------------------------------------------------------------------------------------------

-------------------------------------------------------------------------------------------------------------------------

-------------------------------------------------------------------------------------------------------------------------

5. <u>**How do you describe the typical conditions in the classroom you primarily teach in AUTUMN and SPRING?**</u>

<u>*Temperature in autumn/spring*</u>

*Please tick*

| Uncomfortable | 1 | 2 | 3 | 4 | 5 | Comfortable |
|---|---|---|---|---|---|---|

*Please tick*

| Too hot | 1 | 2 | 3 | 4 | 5 | Too cold |
|---|---|---|---|---|---|---|

*Please tick*

| Stable | 1 | 2 | 3 | 4 | 5 | Varies during the day |
|---|---|---|---|---|---|---|

*Air in autumn/spring*

*Please tick*

| Still | 1 | 2 | 3 | 4 | 5 | Draughty |
|---|---|---|---|---|---|---|

*Please tick*

| Dry | 1 | 2 | 3 | 4 | 5 | Humid |
|---|---|---|---|---|---|---|

*Please tick*

| Fresh | 1 | 2 | 3 | 4 | 5 | Stuffy |
|---|---|---|---|---|---|---|

*Please tick*

| Odourless | 1 | 2 | 3 | 4 | 5 | Smelly |
|---|---|---|---|---|---|---|

*Conditions in autumn/spring*

*Please tick*

| Unsatisfactory overall | 1 | 2 | 3 | 4 | 5 | Satisfactory overall |
|---|---|---|---|---|---|---|

6.    *Please add any additional comments about conditions in autumn and spring in relation to temperature, air ventilation and related indoor environment conditions in the classroom in autumn and spring*

--------------------------------------------------------------------------------------------

--------------------------------------------------------------------------------------------

**Actions to change the temperature & ventilation** <u>in the classroom where you primarily teach</u>

7.    *What action, if any, would you take if you were too warm in the classroom you teach? (i.e., open window, wear lighter clothes)*

--------------------------------------------------------------------------------------------

--------------------------------------------------------------------------------------------

8.    *What action, if any, would you take if you were too cool in the classroom you teach? (i.e., radiator, jumper on)*

--------------------------------------------------------------------------------------------

--------------------------------------------------------------------------------------------

--------------------------------------------------------------------------------------------

9.    *What action, if any, would you take if the classroom you teach needs more ventilation? (i.e., open window, fan)*

--------------------------------------------------------------------------------------------

--------------------------------------------------------------------------------------------

--------------------------------------------------------------------------------------------

10.    *What types of controls are available for you to change the heating, cooling and ventilation in the classroom you teach? (ie radiator, temperature setting, use of windows, etc)*

--------------------------------------------------------------------------------------------

--------------------------------------------------------------------------------------------

--------------------------------------------------------------------------------------------

11.    *In general terms, do you prefer to keep the spaces where you are cool or warm; airy or non-airy? Ie what do you prefer in your house: warm or cool; airy or non-airy?*

--------------------------------------------------------------------------------------------

--------------------------------------------------------------------------------------------

12.    *Do the pupils in your classroom seem comfortable with the indoor conditions of the classroom throughout the year? Do you notice any variations in their comfort in the classroom in different seasons?*

--------------------------------------------------------------------------------------------

--------------------------------------------------------------------------------------------

**Thank you for answering these questions**

**Appendix B. School Adaptations Teacher Survey**

# School adaptations—better indoor environment in teaching/learning spaces

This project by the Welsh School of Architecture investigates actions taken by schools response to health and safety guidance post COVID-19 to promote good indoor environment quality. We want to identify the changes in the use of school buildings including (1) spatial adaptation in classrooms; and, (2) changes in everyday operation and use of building controls in classrooms. Responses will not be associated to specific schools or respondents. Personal information or emails will not be collected. If you have any questions about the study, please contact Dr Gabriela Zapata-Lancaster at ZapataG@cardiff.ac.uk.

**I consent to voluntary participate in this study**. I am aware that data will be kept anonymous and will be held securely by the Cardiff University team (Dr Gabriela Zapata-Lancaster, Dr Thomas Smith, Mr Miltiadis Ionas). I know that if I have any questions about the study, I can email Dr G Zapata-Lancaster at ZapataG@cardiff.ac.uk.

YES            NO

1.   **Date & time**\_\_\_\_\_\_\_\_\_\_\_\_\_\_\_\_\_\_\_\_\_\_\_\_
2.   **Year group you teach** \_\_\_\_\_\_\_\_\_\_\_\_\_\_\_\_
3.   **What are the main changes currently in place in the classroom where you primarily teach adopted in response to COVID-19? Tick all that apply**
     o   Increased hours of outdoor learning
     o   Increased breaks while teaching indoors
     o   Increased opening of windows/doors for more ventilation
     o   Reduced number of students in classroom
     o   New layout- ie distances between desks/seats
     o   Use of protective screens
     o   Others

If others, please give examples

\_\_\_\_\_\_\_\_\_\_\_\_\_\_\_\_\_\_\_\_\_\_\_\_\_\_\_\_\_\_\_\_\_\_\_\_\_\_\_\_\_\_\_\_\_\_\_\_\_\_\_\_\_\_\_\_\_\_\_\_\_\_\_\_\_\_\_

\_\_\_\_\_\_\_\_\_\_\_\_\_\_\_\_\_\_\_\_\_\_\_\_\_\_\_\_\_\_\_\_\_\_\_\_\_\_\_\_\_\_\_\_\_\_\_\_\_\_\_\_\_\_\_\_\_\_\_\_\_\_\_\_\_\_\_

\_\_\_\_\_\_\_\_\_\_\_\_\_\_\_\_\_\_\_\_\_\_\_\_\_\_\_\_\_\_\_\_\_\_\_\_\_\_\_\_\_\_\_\_\_\_\_\_\_\_\_\_\_\_\_\_\_\_\_\_\_\_\_\_\_\_\_

4.   **What are the main changes you have adopted in the classroom where you primarily teach to modify its air quality? (Tick all that apply)**
     o   Frequency/duration of open windows
     o   Frequency/duration of open doors
     o   Others

If others, please give examples

\_\_\_\_\_\_\_\_\_\_\_\_\_\_\_\_\_\_\_\_\_\_\_\_\_\_\_\_\_\_\_\_\_\_\_\_\_\_\_\_\_\_\_\_\_\_\_\_\_\_\_\_\_\_\_\_\_\_\_\_\_\_\_\_\_\_\_

\_\_\_\_\_\_\_\_\_\_\_\_\_\_\_\_\_\_\_\_\_\_\_\_\_\_\_\_\_\_\_\_\_\_\_\_\_\_\_\_\_\_\_\_\_\_\_\_\_\_\_\_\_\_\_\_\_\_\_\_\_\_\_\_\_\_\_

\_\_\_\_\_\_\_\_\_\_\_\_\_\_\_\_\_\_\_\_\_\_\_\_\_\_\_\_\_\_\_\_\_\_\_\_\_\_\_\_\_\_\_\_\_\_\_\_\_\_\_\_\_\_\_\_\_\_\_\_\_\_\_\_\_\_\_

5.  **What changes you have adopted in the classroom where you primarily teach to modify its temperature? (Tick all that apply)**
    - o   Use of radiator valves
    - o   Use of classroom thermostat
    - o   I cannot modify the temperature in the classroom using mechanical systems
    - o   Others

If others, please give examples

______________________________________________________________________

______________________________________________________________________

______________________________________________________________________

6.  **Have you received any advice as to how to use the classroom and/or school spaces to maintain good air quality**

YES                NO

7.  **Could you please tell us any other actions taken (or that could be taken) by the school or in your classroom to improve ventilation/indoor environment in your classrooms? They could be related to adaptation of spaces, use of controls and other relevant actions that have not been included in this questionnaire**

______________________________________________________________________

______________________________________________________________________

______________________________________________________________________

8.  **Could you please tell us if the curriculum/pedagogy of your class enables for opportunities of flexible use of teaching/learning spaces so your pupils can flexibly use different types of spaces to mix indoors and outdoor space use? Ie. Outdoor learning, PE, etc Please give few examples.**

______________________________________________________________________

______________________________________________________________________

______________________________________________________________________

9.  **Please add any additional comments you have related to use of teaching spaces & adaptations to improve indoor environment/air quality in schools**

______________________________________________________________________

______________________________________________________________________

## Appendix C. Comfort in Schools: Pupil Survey

**COMFORT IN SCHOOL SURVEY 2.0**

**Year group** _________________ **Date_____________**

1. **How is your classroom right now?** *Mark x, one response for each topic (temperature, humidity, air freshness)*

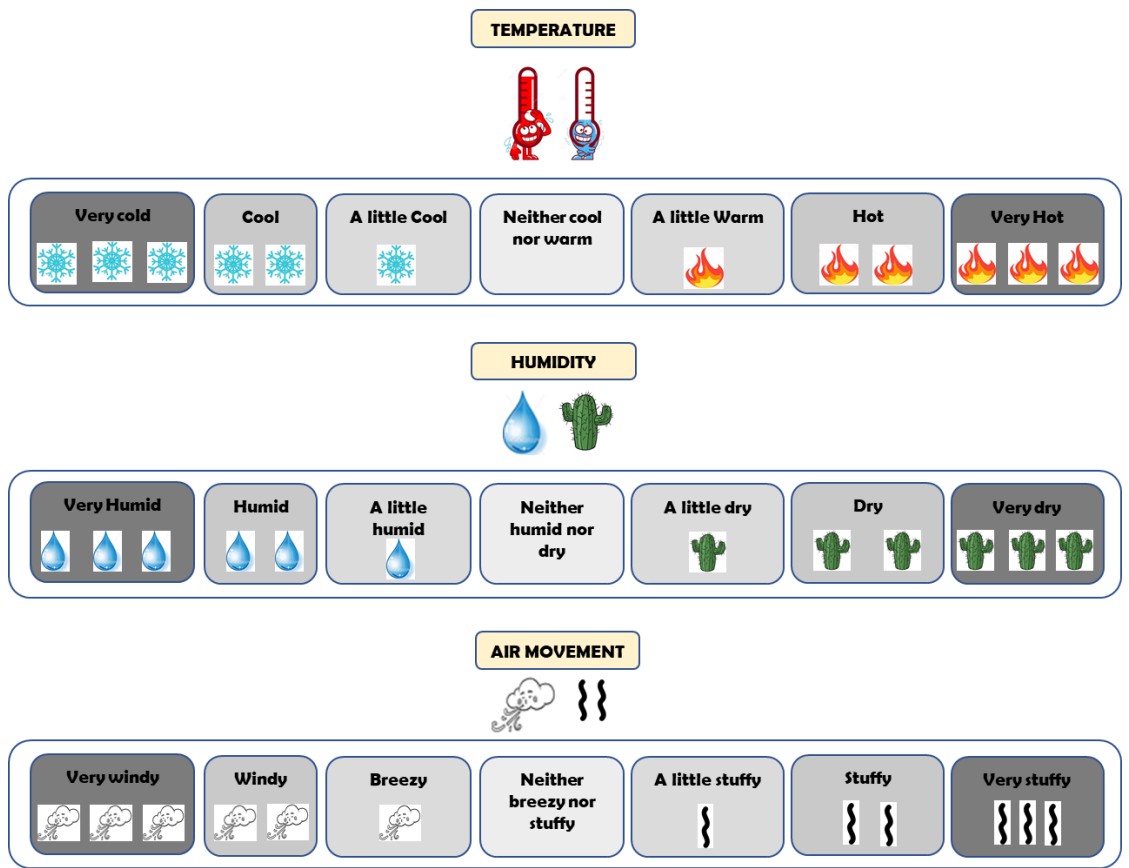

2. **Your classroom makes you feel comfortable or uncomfortable?** *Mark x, one response for each topic (temperature, humidity, air freshness)*

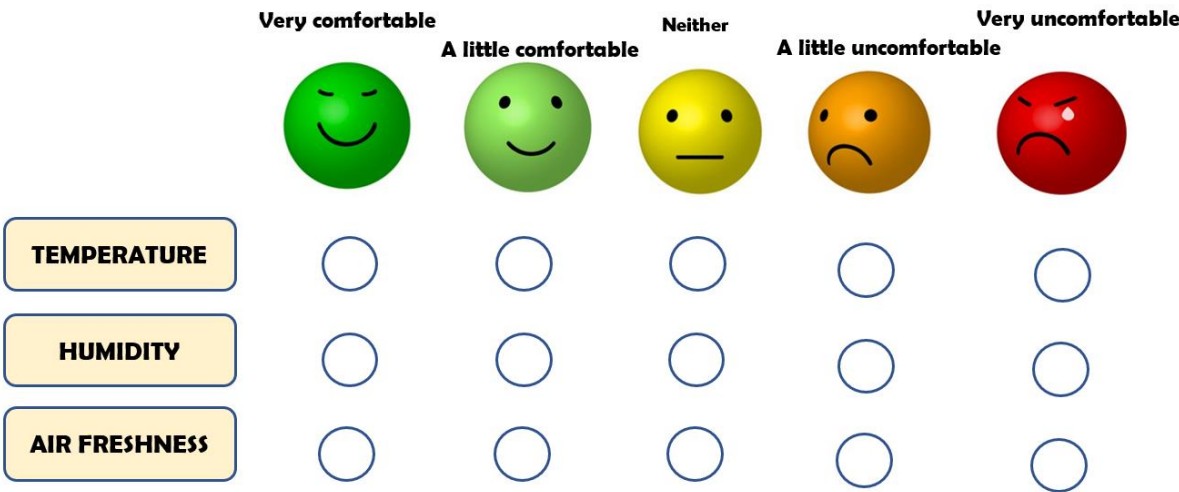



3. **Would you like to change the temperature of your classroom?** *Circle your response*

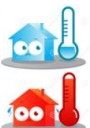

*Right now, I want the temperature of my classroom to be*

    **Warmer**           **No Change, I Like It as It Is**          **Cooler**

                                       **Now**

4. **Would you like to change the freshness of air of your classroom?** *Circle your response*

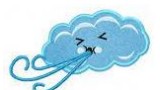

*Right now, I want the air of my classroom to be*

  **Fresher, More Air**           **No Change, I Like It as It Is**       **Less Air Movement,**

                                  **Now**               **Less Breeze**

5. **What clothes are you wearing right now?** *Circle your response*

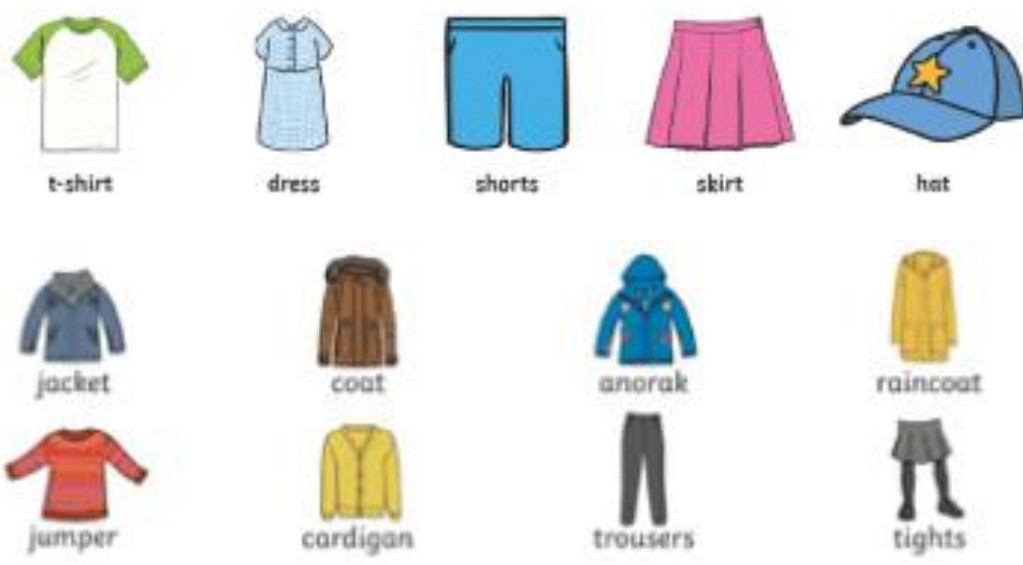

*Thank you!*

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
