# Peer review of "Carbon Dioxide Concentration Levels and Thermal Comfort in Primary School Classrooms: What Pupils and Teachers Do"

_sustainability, doi:10.3390/su15064803_

Round 1

Reviewer 1 Report

Dear Author,

Please report an example of questionnaire, also as annex.

Your file don't report any figure or table, they must be insert in your file. 

Please follow article template. It's very hard to understand your results.

Regards

Author Response

The authors thank the reviewers for their feedback. Please see attachment addressing reviewers' points.

Reviewer 2 Report

In the article entitled “CO2 concentration levels and thermal comfort in primary 2 school classrooms: what pupils and teachers do” the authors analyze the seasonal variability of temperature, CO2 concentration and student satisfaction in two schools – a case study. The subject taken up by the authors in the times of searching for energy efficiency of buildings and problems with air quality is up-to-date and all the results obtained (especially in schools) are useful for building new strategies for optimizing energy demand in the context of ventilation planning. The number of presented results and the number of surveys conducted is sufficient, however, in my opinion the authors did not make satisfactory use of these results to determine new and specific relationships between the variables studied, which means that the general result is general.

My questions to the authors are as follows:

-why do the authors suggest/focus that temperature alone should be the determinant of thermal comfort? Thermal comfort models (for example Fanger based - ISO 7730) take into account other variable parameters: e.g. humidity (measured but not used by the authors), user (kids) activity and what they are wearing (clo)…

- could the survey results be compared with the PMV/PPD ASHRAE-55 and EN 16798 method? And why would the results be different…?

-do the results of the survey for teachers give similar results as for children? Is there a convergence?

- is the questionnaire among teachers statistically representative (what was n=?)?

-were there any additional boundary conditions for the survey? For example, whether it was possible to have a hot meal, a hot drink before tests, or whether children who “felt sick” were eliminated...?

-can you specify the uncertainty of the survey results in percent? (In my opinion, this uncertainty is greater than 25%...) How much is it according to authors?

-How to interpret at temperatures well below the recommended - "very hot" responses of children. In my opinion, parents dressed some children much thicker and hence overheating was detected.... What do the authors think?

-at what point in the measurements were the surveys made, do they concern any specific conditions in the classroom at a given time (as I understand it to be so)  How can the immediate result (at the moment) be related to the long-term result?

-It is not entirely clear what the surveys themselves looked like, the text mentions a 5-point scale and the children, as shown in the picture, had a 3-point scale (cold, ok, warm)? In the text there are descriptions for the children's survey “very hot and very cold” - which indicates 5 degrees...

 - what the conditions of the external/outdoor environment were, i.e. whether opening the windows causes a decrease in CO2 concentration unequivocally in the urban environment? There were no any sources of fuel CO2 emissions?

- nothing is known about the building's technical heating and ventilation systems, it would be useful to supplement it, could such systems affect the obtained results, e.g. dynamic reaction of the system to actual CO2 concentration?

-how to interpret the concentration of CO2 at the level of 3000ppm?

-as a scientific reader, I would like to know if the authors found a correlation between the survey results, temperature and CO2 concentration. Are there any interesting (new) trend/dependency lines, or do both buildings (A and B) give the same trends?

 -as the results show, it can be seen that the temperature in schools is usually too low for most of the measurement period (although the authors write that it is acceptable) and there are even extreme values as 12OC. Maybe authors may consider that at these low temperatures children are much more likely to catch a cold due to temperature and not the COVID infection...

- the authors in numerous verses mention Covid ... how the results relate to the risk of virus infection. Will increasing the number of exchanges by even 0.5 exchanges per hour (lowering already low temperature)  decrease such risk? Can CO2 be a determinant of air quality in the context of COVID risk?

- the article requires an editorial correction, there are typos and an incorrect citation system without brackets

- the introduction is too general and also the results

The whole artcile needs improvement, especially better use of the results to build original conclusions.

Author Response

(The authors gave the same response as above.)

Reviewer 3 Report

Though limited in the number of classrooms/schools monitored, this is generally a great study on CO2 and thermal comfort assessment in classrooms during the COVID-19 pandemic.

It is great to see the conclusion that “the indoor conditions of the classrooms as regards temperature and CO2 were acceptable most of the year and did not prompt serious concerns about the overall indoor quality”, which is consistent with other recent studies on this subject.

However, there are many areas that this paper will benefit from improvement, which are:

  1. The abstract should be more specific on the number of classrooms monitored in the two schools.
  2. In light of the recent COVID-19 pandemic, are CO2 levels in the ranges described in lines 60 to 63 of the introduction section still an indicative benchmark of good indoor air quality? If not, discuss.
  3. Consider explaining what calibration measures were taken into account as regards the reliability of the ALTA sensors' readings.
  4. The methodology section seems to be lacking key information’s and should be improved to cover:

·       The classroom building characteristics such as the length, width and area of the spaces, the type of ventilation available (cross or single sided ventilation? including window configuration), building orientation, and the type of heating systems used

·       The number of occupants in the classrooms and the occupied hours in these schools

·       Images of the classroom, including the location/height of the temperature and CO2 sensors

·       Number of sensors located in each space and decisions on whether the number of sensors/location will be representative of the CO2 and thermal conditions of the monitored spaces

·       The monitoring interval and whether there was any data quality issues.

  1. In line 237, the opening statement in paragraph one of the result section appears incomplete.
  2. In line 101, what is the recommended temperature by CIBSE Guide A guidance and Building Bulletin? The recommended levels should firstly be explained to provide clarity on the criteria used to triage good or poor CO2 and temperature levels.
  3. In the result section, consider whether analysing the presentation of data with combined temperature and CO2 levels within the recommended good levels will provide insightful findings. I.e., X% both temperature and CO2, Y% Temperature but not CO2, Z% CO2 but not temperature and Q% neither CO2 nor temp?
  4. In the conclusion section, the limitations of the study should be well explained.
  5. Consider checking for typographical errors. For example:

·       In line 47 and 48, should it be ‘than’ and not ‘that’?

·       Check lines 91 to 92 of the introduction section, and line 170 of the methodology section

·       Ensure that the correct CO2 subscript symbol is used throughout the document (see lines 108, 138, 151, 159, and 412-415, respectively)

Overall, a very promising paper showing the use of good creative methods, such as visual cues, color-coding and the use of drawings where pupils expressed their perception of the indoor environment in their classrooms and actions they take to modify their thermal experience.

Author Response

(The authors gave the same response as above.)

Round 2

Reviewer 1 Report

Accepted

Reviewer 3 Report

A very brilliant paper showing the use of good creative methods, such as visual cues, colour-coding and the use of drawings where pupils expressed their perception of the indoor environment in their classrooms and actions they take to modify their thermal experience.

Well done for effecting the recommended corrections which has further strengthened your paper.